# Towards Better Selective Classification

**Leo Feng**
Mila – Université de Montréal & Borealis AI
`leo.feng@mila.quebec`

**Mohamed Osama Ahmed**
Borealis AI
`mohamed.o.ahmed@borealisai.com`

**Hossein Hajimirsadeghi**
Borealis AI
`hossein.hajimirsadeghi@borealisai.com`

**Amir Abdi**
Borealis AI
`amir.abdi@borealisai.com`

## Abstract

We tackle the problem of Selective Classification where the objective is to achieve the best performance on a predetermined ratio (coverage) of the dataset. Recent state-of-the-art selective methods come with architectural changes either via introducing a separate selection head or an extra abstention logit. In this paper, we challenge the aforementioned methods. The results suggest that the superior performance of state-of-the-art methods is owed to training a more generalizable classifier rather than their proposed selection mechanisms. We argue that the best performing selection mechanism should instead be rooted in the classifier itself. Our proposed selection strategy uses the classification scores and achieves better results by a significant margin, consistently, across all coverages and all datasets, without any added compute cost. Furthermore, inspired by semi-supervised learning, we propose an entropy-based regularizer that improves the performance of selective classification methods. Our proposed selection mechanism with the proposed entropy-based regularizer achieves new state-of-the-art results.

## 1 Introduction

A model's ability to abstain from a decision when lacking confidence is essential in mission-critical applications. This is known as the Selective Prediction problem setting. The abstained and uncertain samples can be flagged and passed to a human expert for manual assessment, which, in turn, can improve the re-training process. This is crucial in problem settings where confidence is critical or an incorrect prediction can have significant consequences such as in the financial, medical, or autonomous driving domains. Several papers have tried to address this problem by estimating the uncertainty in the prediction. Gal & Ghahramani (2016) proposed using MC-dropout. Lakshminarayanan et al. (2017) proposed to use an ensemble of models. Dusenberry et al. (2020) and Maddox et al. (2019) are examples of work using Bayesian deep learning. These methods, however, are either expensive to train or require lots of tuning for acceptable results.

In this paper, we focus on the Selective Classification problem setting where a classifier has the option to abstain from making predictions. Models that come with an abstention option and tackle the selective prediction problem setting are naturally called *selective models*. Different selection approaches have been suggested such as incorporating a selection head Geifman & El-Yaniv (2019) or an abstention logit (Huang et al., 2020; Ziyin et al., 2019). In either case, a threshold is set such that selection and abstention values above or below the threshold decide the selection action. SelectiveNet Geifman & El-Yaniv (2019) proposes to learn a model comprising of a selection head and a prediction head where the values returned by the selection head determines whether the datapoint is selected for prediction or not. Huang et al. (2020) and Ziyin et al. (2019) introduced an additional abstention logit for classification settings where the output of the additional logit determines whether the model abstains from making predictions on the sample. The promising results of these works suggest that the selection mechanism should focus on the output of an external head/logit.

On the contrary, in this work, we argue that the selection mechanism should be rooted in the classifier itself. The results of our rigorously conducted experiments show that (1) the superior

performance of the state-of-the-art methods is owed to training a more generalizable classifier rather than their proposed external head/logit selection mechanisms. These results suggest that future work in selective classification (i) should aim to learn a more generalizable classifier and (ii) the selection mechanism should be based on the classifier itself rather than the recent research directions of architecture modifications for an external logit/head. (2) We highlight a connection between selective classification and semi-supervised learning. To the best of our knowledge, this has has not been explored before. We show that entropy-minimization regularization, a common technique in semi-supervised learning, significantly improves the performance of the state-of-the-art selective classification method. The promising results suggest that additional research is warranted to explore the relationship between these two research directions.

From a practical perspective, (3) we propose a selection mechanism that outperforms the original selection mechanism of state-of-the-art methods. Furthermore, this method can be immediately applied to an already deployed selective classification model and instantly improve performance at no additional cost. (4) We show a selective classifier trained with the entropy-regularised loss and with selection according to the classification scores achieves new state-of-the-art results by a significant margin (up to $80\%$ relative improvement). (5) Going beyond the already-saturated datasets often used for Selective Classification research, we include results on larger datasets: StanfordCars, Food101, Imagenet, and Imagenet100 to test the methods on a wide range of coverages and ImagenetSubset to test the scalability of the methods.

## 2 RELATED WORK

The option to reject a prediction has been explored in depth in various learning algorithms not limited to neural networks. Primarily, Chow (Chow, 1970) introduced a cost-based rejection model and analysed the error-reject trade-off. There has been significant study in rejection in Support Vector Machines (Bartlett & Wegkamp, 2008; Fumera & Roli, 2002; Wegkamp, 2007; Wegkamp & Yuan, 2011). The same is true for nearest neighbours (Hellman, 1970) and boosting (Cortes et al., 2016).

In 1989, LeCun et al. (1989) proposed a rejection strategy for neural networks based on the most activated output logit, second most activated output logit, and the difference between the activated output logits. Geifman & El-Yaniv (2017) presented a technique to achieve a target risk with a certain probability for a given confidence-rate function. As examples of confidence-rate functions, the authors suggested selecting according to Softmax Response and MC-Dropout as selection mechanisms for a vanilla classifier. We build on this idea to demonstrate that Softmax Response, if utilized correctly, is the highest performing selection mechanism in the selective classification settings. Beyond selective classification, max-logit (Softmax Response) has also been used in anomaly detection (Hendrycks & Gimpel, 2016; Dietterich & Guyer, 2022).

Future work focused on architectural changes and selecting according to a separately computed head/logit with their own parameters. The same authors, Geifman & El-Yaniv (2019) later proposed SelectiveNet (see Section 3.2.1), a three-headed model, comprising of heads for selection, prediction, and auxiliary prediction. Deep Gamblers (Ziyin et al., 2019) (see Appendix A.1) and Self-Adaptive Training (Huang et al., 2020) (see Section 3.3.1) propose a $(C + 1)$-way classifier, where $C$ is the number of classes and the additional logit represents abstention. In contrast, in this work, we explain how selecting via entropy and max-logit can work as a proxy to select samples which could potentially minimise the cross entropy loss. In general, we report the surprising results that the selection head of the SelectiveNet and the abstention logits in Deep Gamblers and Self-Adaptive Training are suboptimal selection mechanisms. Furthermore, their previously reported good performance is rooted in their optimization process converging to a more generalizable model.

Another line of work which tackles the selective classification is that of cost-sensitive classification (Charoenphakdee et al., 2021). However, the introduction of the target coverage adds a new variable and changes the mathematical formulation. Other works have proposed to perform classification in conjunction with expert decision makers (Mozannar & Sontag, 2020).

In this work, we also highlight a connection between semi-supervised learning and selective classification, which, to the best of our knowledge, has not been explored before. As a result, we propose an entropy-regularized loss function in the Selective Classification settings to further improve the performance of the Softmax Response selection mechanism. However, entropy minimization

objectives have been widely used for Unsupervised Learning (Long et al., 2016), Semi-Supervised Learning (Grandvalet & Bengio, 2004), and Domain Adaptation (Vu et al., 2019; Wu et al., 2020).

## 3 BACKGROUND

In this section, we introduce the Selective Classification problem. Additionally, we describe the top methods for Selective Classification. To the best of our knowledge, Self-Adaptive Training (Huang et al., 2020) achieves the best performance on the selective classification datatests.

### 3.1 PROBLEM SETTING: SELECTIVE CLASSIFICATION

The selective prediction task can be formulated as follows. Let $\mathcal{X}$ be the feature space, $\mathcal{Y}$ be the label space, and $P(\mathcal{X}, \mathcal{Y})$ represent the data distribution over $\mathcal{X} \times \mathcal{Y}$. A selective model comprises of a prediction function $f : \mathcal{X} \rightarrow \mathcal{Y}$ and a selection function $g : \mathcal{X} \rightarrow \{0, 1\}$. The selective model decides to make predictions when $g(x) = 1$ and abstains from making predictions when $g(x) = 0$. The objective is to maximise the model's predictive performance for a given target coverage $c_{\text{target}} \in [0, 1]$, where coverage is the proportion of the selected samples. The selected set is defined as $\{x : g(x) = 1\}$. Formally, an optimal selective model, parameterised by $\theta^*$ and $\psi^*$, would be the following:

$$\theta^*, \psi^* = \text{argmin}_{\theta, \psi} \mathbb{E}_P[l(f_\theta(x), y) \cdot g_\psi(x)], \quad \text{s.t. } \mathbb{E}_P[g_\psi(x)] \geq c_{\text{target}}, \tag{1}$$

where $\mathbb{E}_P[l(f_\theta(x), y) \cdot g_\psi(x)]$ is the selective risk. Naturally, higher coverages are correlated with higher selective risks.

In practice, instead of a hard selection function $g_\psi(x)$, existing methods aim to learn a *soft* selection function $\bar{g}_\psi : \mathcal{X} \rightarrow \mathbb{R}$ such that larger values of $\bar{g}_\psi(x)$ indicate the datapoint should be selected for prediction. At test time, a threshold $\tau$ is selected for a coverage $c$ such that

$$g_\psi(x) = \begin{cases} 1 & \text{if } \bar{g}_\psi(x) \geq \tau \\ 0 & \text{otherwise} \end{cases}, \quad \text{s.t. } \mathbb{E}[g_\psi(x)] = c_{\text{target}} \tag{2}$$

In this setting, the selected (covered) dataset is defined as $\{x : \bar{g}_\psi(x) \geq \tau\}$. The process of selecting the threshold $\tau$ is known as *calibration*.

### 3.2 APPROACH: LEARN TO SELECT

#### 3.2.1 SELECTIVENET

SelectiveNet (Geifman & El-Yaniv, 2019) is a three-headed network proposed for selective learning. A SelectiveNet model has three output heads designed for selection $\bar{g}$, prediction $f$, and auxiliary prediction $h$. The selection head infers the selective score of each sample, as a value between 0 to 1, and is implemented with a sigmoid activation function. The auxiliary prediction head is trained with a standard (non-selective) loss function. Given a batch $\{(x_i, y_i)\}_{i=1}^m$, where $y_i$ is the label, the model is trained to minimise the loss $\mathcal{L}$ where it is defined as:

$$\mathcal{L} = \alpha \left( \mathcal{L}_{selective} + \lambda \mathcal{L}_c \right) + (1 - \alpha) \mathcal{L}_{aux}, \tag{3}$$

$$\mathcal{L}_{selective} = \frac{\frac{1}{m} \sum_{i=1}^m \ell(f(x_i), y_i) \bar{g}(x_i)}{\frac{1}{m} \sum_{i=1}^m \bar{g}(x_i)}, \tag{4}$$

$$\mathcal{L}_c = \max(0, (c_{\text{target}} - \frac{1}{m} \sum_{i=1}^m \bar{g}(x_i))^2), \quad \mathcal{L}_{aux} = \frac{1}{m} \sum_{i=1}^m \ell(h(x_i), y_i), \tag{5}$$

where $\ell$ is any standard loss function. In Selective Classification, $\ell$ is the Cross Entropy loss function. The coverage loss $\mathcal{L}_c$ encourages the model to achieve the desired coverage and ensures $\bar{g}(x_i) > 0$ for at least $c_{\text{target}}$ proportion of the batch samples. The selective loss $L_{selective}$ discounts the weight of difficult samples via the soft selection value $\bar{g}(x)$ term encouraging the model to focus more on easier samples which the model is more confident about.

The auxiliary loss $L_{aux}$ ensures that all samples, regardless of their selective score ($\bar{g}(x)$), contribute to the learning of the feature model. $\lambda$ and $\alpha$ are hyper-parameters controlling the trade-off of

different terms. Unlike Deep Gamblers and Self-Adaptive Training, SelectiveNet trains a separate model for each target coverage $c_{\text{target}}$. In the SelectiveNet paper (Geifman & El-Yaniv, 2019), it has been suggested that the best performance is achieved when the training target coverage is equal to that of the evaluation coverage.

### 3.3 APPROACH: LEARN TO ABSTAIN

Self-Adaptive Training (Huang et al., 2020) and Deep Gamblers (Ziyin et al., 2019) propose to tackle the selective classification problem by introducing a $(C + 1)$-th class logit where the extra class logit represents abstention. Let $p_\theta(\cdot|x)$ represent the prediction network with softmax as the last layer. This family of methods abstain if $p_\theta(C + 1|x)$ is above a threshold. Here, we unify the notations of these abstention and selection methods under the same Selective Classification framework (See Section 3.1) with the following soft selection function: $\bar{g}(x) = 1 - p_\theta(C + 1|x)$. Due to the space limitation, the formulation for Deep Gamblers is included in the Appendix.

#### 3.3.1 SELF-ADAPTIVE TRAINING

In addition to learning a logit that represents abstention, Self-Adaptive Training (Huang et al., 2020) proposes to use a convex combination of labels and predictions as a dynamically moving training target instead of the fixed labels. Let $\mathbf{y}_i$ be the one-hot encoded vector representing of the label for a datapoint $(x_i, y_i)$ where $y_i$ is the label.

Initially, the model is trained with a cross-entropy loss for a series of pre-training steps. Afterwards, the model is updated according to a dynamically moving training target. The training target $t_i$ is initially set equal to the label $t_i \leftarrow \mathbf{y}_i$ such that the training target is updated according to $t_i \leftarrow \alpha \times t_i + (1 - \alpha) \times p_\theta(\cdot|x_i)$ s.t. $\alpha \in (0, 1)$ after each model update. Similar to Deep Gamblers, the model is trained to optimise a loss function that allows the model to also choose to abstain on hard samples instead of making a prediction:

$$\mathcal{L} = -\frac{1}{m} \sum_{i=1}^{m} [t_{i,y_i} \log p_\theta(y_i|x_i) + (1 - t_{i,y_i}) \log p_\theta(C + 1|x_i)], \tag{6}$$

where $m$ is the number of datapoints in the batch. As training progresses, $t_i$ approaches $p_\theta(\cdot|x_i)$. The first term is similar to the Cross Entropy Loss and encourages the model to learn a good classifier. The second term encourages the model to abstain from making predictions for samples that the model is uncertain about. This use of dynamically moving training target $t_i$ allows the model to avoid fitting on difficult samples as the training progresses.

## 4 METHODOLOGY

We motivate an alternative selection mechanism Softmax Response for Selective Classification models. We explain how the known state-of-the-art selective methods can be equipped with the proposed selection mechanism and why it further improves performance. Inspired by semi-supervised learning, we also introduce an entropy-regularized loss function.

### 4.1 MOTIVATION

Recent state-of-the-art methods have proposed to learn selective models with architecture modifications such as an external logit/head. These architecture modifications, however, act as regularization mechanisms that allow the method to train more generalizable classifiers (see Table 1). As a result, the claimed improved results from these models could actually be attributed to their classifiers being more generalizable. For these selective models to have strong performance in selective classification they require the external logit/head to generalise in these sense that the external logit/head must select samples for which the classifier is confident of its prediction. Since the logit/head has its own set of learned model parameters, this adds another potential mode of failure for a selective model. Specifically, the learned parameters can fail to generalise and the logit/head may (1) suggest samples for which the classifier is not confident about and (2) reject samples for which the classifier is confident about. In the appendix (See Figure 4 and 5), we include examples of images that fail due

| Model | Accuracy |
|---|---|
| Vanilla Classifier | 85.68 ± 0.14 |
| SelectiveNet | 86.23 ± 0.14 |
| Deep Gamblers | 86.51 ± 0.52 |
| Self-Adaptive Training | 86.40 ± 0.30 |

Table 1: Results at 100% coverage on Imagenet100.

to this introduced mode of failure. As such, we propose that selective mechanisms should stem from the classifier itself instead of an external logit/head, avoiding this extra mode of failure.

## 4.2 SELECTING ACCORDING TO THE CLASSIFIER

The cross entropy loss function is a popular loss function for classification due to its differentiability. However, during evaluation, the most utilized metric is accuracy, *i.e.*, whether a datapoint is predicted correctly. In the cross-entropy objective of the conventional classification settings, $p(c|x_i)$ is a one-hot encoded vector; therefore, the the cross-entropy loss can be simplified as $CE\left(p(\cdot|x_i), p_\theta(\cdot|x_i)\right) = -\sum_{u=1}^{C} p(u|x_i) \log p_\theta(u|x_i) = -\log p_\theta(y_i|x_i)$, *i.e.*, during optimization, the logit of the correct class is maximised. Accordingly, the maximum value of logits can be interpreted as the model's relative confidence of its prediction. Therefore, a simple selection mechanism for a model would be to select according to the maximum predictive class score, $\bar{g}(x) = \max_{u \in \{1,...C\}} p_\theta(u|x_i)$ (aka Softmax Response (Geifman & El-Yaniv, 2017)). Alternatively, a model can also select according to its predictive entropy $\bar{g}(x) = -H(p_\theta(\cdot|x))$, a metric of the model's uncertainty. An in-depth discussion is included in in the Appendix B.

## 4.3 RECIPE FOR BETTER SELECTIVE CLASSIFICATION

The recipe that we are providing for better selective classification is as follows:

1. Train a selective classifier (*e.g.*, SelectiveNet, Self-Adaptive Training, or Deep Gamblers).
2. Discard its selection mechanism:
   - For SelectiveNet: Ignore the selection head
   - For Self-Adaptive Training and Deep Gambler: Ignore the additional abstain logit and compute the final layer's softmax on the original $C$ class logits.
3. Use a classifier-based selection mechanism (e.g., Softmax Response) to rank the samples.
4. Calculate the threshold value $\tau$, based on the validation set, to achieve the desired target coverage and select samples with max logit greater than $\tau$.

Empirically, we show that selecting via entropy or Softmax Response both outperform selecting according to the external head/logit. From these results, we can conclude that the strong performance of these recent state-of-the-art methods were due to learning a more generalizable classifier rather than their proposed selection mechanisms. In Step 3, we experimented with both an entropy-based selection mechanism and Softmax Response but we found that Softmax Response performed better. Notably, Softmax Response does not require retraining and can be immediately applied to already deployed models for significant performance improvement at negligible cost.

## 4.4 ENTROPY-REGULARIZED LOSS FUNCTION

Here, we highlight a similarity between semi-supervised learning and selective classification, which to our knowledge has not been explored before. In the semi-supervised learning setting the training dataset consists of labelled and unlabelled data. A simple approach is to train solely on the labelled data and ignore the unlabelled data, i.e., training a model via supervised learning to the labelled data. This is equivalent to having a weight of 1 for the labelled samples and 0 for the unlabelled samples. However, this is suboptimal because it does not use any information from the unlabelled samples. Similarly, in Selective Classification, samples that are selected tend to have a high weight close to 1 (see, for example, in Section 3.2.1, the $\bar{g}(x)$ term in the objective) and samples that are not selected

have a low weight close to 0. One way that semi-supervised learning have proposed to tackle this is via an entropy minimization term.

Entropy minimization is one of the most standard, well-studied, and intuitive methods for semi-supervised learning. It uses the information of all the samples and increases the model's confidence in its predictions, including on the unlabelled samples, resulting in a better classifier. Inspired by the similarity in the objective of selective classification and the setting of semi-supervised learning, we propose an entropy-minimisation term for the objective function of selective classification methods:

$$\mathcal{L}_{new} = \mathcal{L} + \beta \, \mathcal{H}(p_\theta(\cdot|x)), \tag{7}$$

where $\beta$ is a hyperparameter that controls the impact. In our experiments, we found $\beta = 0.01$ to perform well in practice. The entropy minimization term encourages the model to be more confident in its predictions, *i.e.*, increasing the confidence of the predicted class and decreasing the predictive entropy during training. Thus, it allows for better disambiguation between sample predictions. The larger coefficient on the cross-entropy term compared to that of the entropy-minimization term ensures that increasing the confidence of correct predictions are prioritised, benefitting Softmax Response. In Section 5, we show that this proposed loss function based on semi-supervised learning improves the performance in Selective Classification by a significant margin. These results opens the door to future exploration of the connection between Selective Classification and semi-supervised learning.

## 5 EXPERIMENTS

For the following experiments, we evaluate the following state-of-the-art methods (1) SelectiveNet (SN), (2) Self-Adaptive Training (SAT), and (3) Deep Gamblers. Furthermore, we compare the performance of these methods with the following selection mechanisms (1) original selection mechanism and (2) SR: Softmax Response (our proposed method). Due to space limitations, the table results for a vanilla classifier is included in the Appendix with several additional results.

The goal of our experimental evaluation is to answer the following questions: (1) Is the superior performance of recent state-of-the-art methods due to their proposed external head/logit selection mechanisms? Taking this further, what is the state-of-the-art selection mechanism? (2) Does the proposed entropy-regularized loss function improve the effectiveness of Softmax Response for Selective Classification? (3) What is the new state-of-the-art method for Selective Classification? (4) How scalable are selective methods for larger datasets with larger number of classes?

### 5.1 DATASETS

We introduce new datasets: StanfordCars, Food101, Imagenet, Imagenet100 and ImagenetSubset, for the selective classification problem setting and benchmark the existing state-of-the-art methods. We propose StanfordCars, Food101, Imagenet, and Imagenet100, as realistic non-saturated datasets that can be evaluated at a wide range of coverages $(10 - 100\%)$. In addition, we propose ImagenetSubset as a collection of datasets to evaluate the scalability of the methods for different number of classes. This is in contrast to the existing Selective Classification research which mainly have focused on small datasets such as CIFAR-10 with 10 or less classes, low resolution images (64x64 or less), and very low error (The error at $80\%$ coverage is already lower than 1%) so this dataset is limited to high coverages $(70\%+)$. The results of the previously introduced datasets indicate saturation, *e.g.*, $0.3\%$ error at 70% coverage, discouraging experiments with lower coverages, which, in turn, prevents researchers from achieving conclusive results.

**Imagenet/Imagenet100/ImagenetSubset.** Imagenet (Deng et al., 2009) comprises of 1,300 images per class and evaluation data comprising of 50,000 images split into 1,000 classes. Imagenet100 (Tian et al., 2020) is a subset of Imagenet which comprising of 100 classes. ImagenetSubset is created as a collection of datasets with varying number of classes (difficulty) from 25 to 175 in increments of 25. The classes are sampled randomly such that datasets with less classes are subsets of those with more classes. The complete list of selected classes in each dataset subset is available in the Appendix[1]. ImagenetSubset evaluates the models' performance with respect to the difficulty (scalability) of the task.

---

[1]Note that the created dataset of ImagenetSubset with 100 classes is different than that of Imagenet100.

| Coverage | SelectiveNet (SN) | | Deep Gamblers (DG) | | Self-Adaptive Training (SAT) | |
|---|---|---|---|---|---|---|
| | SN | SN+SR | DG | DG+SR | SAT | SAT+SR |
| 100 | 13.77 ± 0.14 | 13.77 ± 0.14 | 13.49 ± 0.52 | 13.49 ± 0.52 | 13.58 ± 0.30 | 13.58 ± 0.30 |
| 90 | 9.44 ± 0.28 | 7.89 ± 0.10 | 8.42 ± 0.44 | 8.11 ± 0.48 | 8.80 ± 0.41 | 8.04 ± 0.25 |
| 80 | 6.00 ± 0.22 | 4.47 ± 0.19 | 5.21 ± 0.32 | 4.52 ± 0.38 | 5.20 ± 0.29 | 4.46 ± 0.13 |
| 70 | 3.38 ± 0.21 | 2.21 ± 0.37 | 3.30 ± 0.40 | 2.58 ± 0.21 | 2.71 ± 0.29 | 2.33 ± 0.19 |
| 60 | 1.99 ± 0.15 | 1.57 ± 0.06 | 2.14 ± 0.37 | 1.71 ± 0.32 | 1.72 ± 0.11 | 1.37 ± 0.12 |
| 50 | 1.05 ± 0.17 | 0.85 ± 0.02 | 1.55 ± 0.27 | 1.31 ± 0.22 | 1.18 ± 0.14 | 0.88 ± 0.07 |
| 40 | 0.58 ± 0.08 | 0.53 ± 0.03 | 1.23 ± 0.38 | 1.07 ± 0.19 | 0.82 ± 0.06 | 0.60 ± 0.11 |
| 30 | 1.04 ± 0.37 | 0.64 ± 0.10 | 1.09 ± 0.31 | 0.96 ± 0.21 | 0.67 ± 0.06 | 0.59 ± 0.11 |
| 20 | 48.87 ± 6.15 | 47.10 ± 3.83 | 1.03 ± 0.31 | 0.90 ± 0.22 | 0.48 ± 0.18 | 0.46 ± 0.22 |
| 10 | 99.00 ± 0.00 | 99.00 ± 0.00 | 0.80 ± 0.28 | 0.53 ± 0.25 | 0.32 ± 0.10 | 0.12 ± 0.16 |

Table 2: Comparison of the selective classification error between SelectiveNet (SN), Deep Gamblers (DG), Self-Adaptive Training (SAT) with their original selection mechanisms vs. using Softmax Response (SR) as the selection mechanisms on ImageNet100.

**Food101.** The Food dataset (Bossard et al., 2014) contains 75750 training images and 25250 testing images split into 101 food categories.

**StanfordCars.** The Cars dataset (Krause et al., 2013) contains 8,144 training images and 8,041 testing images split into 196 classes of cars. Unlike prior works which typically evaluate StanfordCars for transfer learning, in this work, the models are trained from scratch.

**CIFAR-10.** The CIFAR-10 dataset (Krizhevsky, 2009) comprises of small images: 50,000 images for training and 10,00 images for evaluation split into 10 classes. Each image is of size $32 \times 32 \times 3$.

## 5.2 EXPERIMENT DETAILS

For our experiments, we adapted the publicly available official implementations of Deep Gamblers and Self-Adaptive Training [2]. Experiments on SelectiveNet were conducted with our Pytorch implementation of the method which follow the details provided in the original paper (Geifman & El-Yaniv, 2019). For the StanfordCars, Food101, Imagenet100, and ImagenetSubset datasets, we use a ResNet34 architecture for Deep Gamblers, Self-Adaptive Training, and the main body block of SelectiveNet. Following prior work, we use a VGG16 architecture for the CIFAR-10 experiments.

We tuned the entropy minimization loss function hyperparameter with the following values: $\beta \in \{0.1, 0.01, 0.001, 0.0001\}$. CIFAR10, Food101, and StanfordCars experiments were run with 5 seeds. Imagenet-related experiments were run with 3 seeds. Additional details regarding hyperparameters are included in the Appendix.

## 5.3 RESULTS

### 5.3.1 CORRECTING THE MISCONCEPTION ABOUT THE SELECTION MECHANISM

In Table 2, we compare the different selection mechanisms for a given selective classification method (SelectiveNet, Deep Gamblers, and Self-Adaptive Training). The results show that for each of these trained selective classifiers, their original selection mechanisms are suboptimal; in fact, selecting via Softmax Response outperforms their original selection mechanism. These results suggest that (1) the strong performance of these methods were due to them learning a more generalizable model rather than their proposed external head/logit selection mechanisms and (2) the selection mechanism should stem from the classifier itself rather than a separate head/logit. We see that Softmax Response is the state-of-the-art selection mechanism. It is important to note that this performance gain is achieved by simply changing the selection mechanism of the pre-trained selective model without any additional computational cost. This observation applies to SN, DG, and SAT models.

An interesting result from this experiment is that at low coverages (30%, 20%, and 10%), SelectiveNet's performance progressively gets worse. We hypothesize that this is due to the optimisation process of SelectiveNet that allows the model to disregard (*i.e.*, assign lower weight to their loss) a vast majority of samples during training at little cost, *i.e.*, $\bar{g}(x) \approx 0$, especially when the target

---

[2]The code is available at https://github.com/BorealisAI/towards-better-sel-cls.

| | StanfordCars | | | | Food101 | | |
|---|---|---|---|---|---|---|---|
| Cov. | SAT | SAT+SR | SAT+EM+SR | | SAT | SAT+SR | SAT+EM+SR |
| 100 | 37.68 ± 1.11 | 37.68 ± 1.11 | **32.49 ± 2.33** | | **16.41 ± 0.10** | **16.41 ± 0.10** | **16.32 ± 0.35** |
| 90 | 32.34 ± 1.19 | 32.04 ± 1.18 | **26.60 ± 2.39** | | 11.87 ± 0.13 | **10.84 ± 0.17** | **10.77 ± 0.36** |
| 80 | 26.86 ± 1.15 | 26.39 ± 1.13 | **20.87 ± 2.33** | | 7.99 ± 0.12 | 6.57 ± 0.13 | **6.57 ± 0.21** |
| 70 | 21.34 ± 1.20 | 20.70 ± 1.23 | **15.84 ± 1.98** | | 4.89 ± 0.11 | **3.52 ± 0.05** | **3.52 ± 0.19** |
| 60 | 16.21 ± 1.10 | 14.92 ± 1.03 | **11.09 ± 1.50** | | 2.73 ± 0.09 | **1.95 ± 0.08** | **1.75 ± 0.17** |
| 50 | 11.59 ± 0.74 | 10.25 ± 0.97 | **7.00 ± 1.13** | | 1.38 ± 0.09 | **1.06 ± 0.06** | **0.96 ± 0.14** |
| 40 | 7.76 ± 0.43 | 6.32 ± 0.69 | **4.00 ± 0.87** | | 0.79 ± 0.05 | **0.56 ± 0.08** | **0.49 ± 0.08** |
| 30 | 4.56 ± 0.35 | 3.54 ± 0.36 | **2.20 ± 0.44** | | 0.48 ± 0.07 | 0.32 ± 0.04 | **0.19 ± 0.03** |
| 20 | 2.42 ± 0.36 | 1.93 ± 0.09 | **1.17 ± 0.28** | | 0.25 ± 0.01 | **0.15 ± 0.01** | **0.09 ± 0.05** |
| 10 | 1.49 ± 0.00 | **1.20 ± 0.21** | **0.80 ± 0.22** | | 0.15 ± 0.07 | 0.09 ± 0.02 | **0.03 ± 0.02** |

Table 3: Comparison of the selective classification error between Self-Adaptive Training (SAT) with the original selection mechanisms vs. using Softmax Response (SR) and the proposed entropy minimization loss function (EM) on StanfordCars and Food101

| | Imagenet | | | Imagenet100 | | | |
|---|---|---|---|---|---|---|---|
| Cov. | SAT | SAT+EM+SR | | SAT | SAT + SR | SAT + EM | SAT+EM+SR |
| 100 | **27.41 ± 0.08** | **27.27 ± 0.05** | | **13.58 ± 0.30** | **13.58 ± 0.30** | **13.18 ± 0.24** | **13.18 ± 0.24** |
| 90 | 22.67 ± 0.24 | **21.57 ± 0.19** | | 8.80 ± 0.41 | **8.04 ± 0.25** | 8.69 ± 0.32 | **7.73 ± 0.22** |
| 80 | 18.14 ± 0.28 | **16.83 ± 0.06** | | 5.20 ± 0.29 | 4.46 ± 0.13 | 5.03 ± 0.36 | **3.90 ± 0.34** |
| 70 | 13.88 ± 0.14 | **12.34 ± 0.11** | | 2.71 ± 0.29 | 2.33 ± 0.19 | 2.61 ± 0.22 | **1.81 ± 0.27** |
| 60 | 10.11 ± 0.15 | **8.45 ± 0.05** | | 1.72 ± 0.11 | 1.37 ± 0.12 | 1.59 ± 0.19 | **0.95 ± 0.13** |
| 50 | 6.82 ± 0.07 | **5.57 ± 0.17** | | 1.18 ± 0.14 | 0.88 ± 0.07 | 1.02 ± 0.21 | **0.62 ± 0.09** |
| 40 | 4.32 ± 0.33 | **3.77 ± 0.00** | | 0.82 ± 0.06 | 0.60 ± 0.11 | 0.81 ± 0.12 | **0.34 ± 0.06** |
| 30 | 2.68 ± 0.14 | **2.32 ± 0.15** | | 0.67 ± 0.06 | 0.59 ± 0.11 | 0.61 ± 0.14 | **0.25 ± 0.10** |
| 20 | 1.82 ± 0.13 | **1.35 ± 0.20** | | 0.48 ± 0.18 | 0.46 ± 0.22 | 0.52 ± 0.16 | **0.15 ± 0.08** |
| 10 | 1.27 ± 0.34 | **0.55 ± 0.05** | | **0.32 ± 0.10** | **0.12 ± 0.16** | **0.32 ± 0.20** | **0.12 ± 0.07** |

Table 4: Results on Imagenet and demonstration of the impact of our SAT+EM+SR method over using SR alone or EM alone.

coverage is as low as 10%. In contrast, Deep Gamblers and Self-Adaptive Training models are equally optimised over all samples regardless of their selection.

### 5.3.2 Power of Softmax Response Selection with Entropy Minimization

In these experiments, we focus on Self-Adaptive Training as it is the state-of-the-art selective model. In Table 3, 4, and 5, we compare Self-Adaptive Training (SAT), SAT with SR (Softmax Response) selection mechanism, and SAT with SR and EM (Entropy-Minimization) on the Imagenet, Stanford-Cars, Food101, and CIFAR10 datasets. The results show that SAT+EM+SR achieves state-of-the-art performance across all coverages. For example, in StanfordCars, at 70% coverage, we see a raw 5.5% absolute improvement (25% relative reduction) in selective classification error by using our proposed

| | CIFAR10 | |
|---|---|---|
| Coverage | SAT | SAT+EM+SR |
| 100 | **5.91 ± 0.04** | **5.91 ± 0.04** |
| 95 | **3.73 ± 0.13** | **3.63 ± 0.10** |
| 90 | **2.18 ± 0.11** | **2.11 ± 0.06** |
| 85 | **1.26 ± 0.09** | **1.18 ± 0.07** |
| 80 | **0.69 ± 0.04** | **0.64 ± 0.04** |
| 75 | **0.37 ± 0.01** | **0.36 ± 0.03** |
| 70 | **0.27 ± 0.02** | **0.23 ± 0.05** |

Table 5: Results on CIFAR10

method EM+SR. In Food101, at 70% coverage, we see a raw 1.37% absolute reduction (28% relative reduction) in selective classification error. The clear and considerable improvement across all coverages when using Softmax Response selection mechanism rather than the original selection mechanism. These results further confirm our surprising finding that existing selection mechanisms are suboptimal. In the Appendix we further include (1) risk-coverage curves and (2) results for several network architectures. The results of those experiments show that our proposed methodology generalises across different network architectures.

For the CIFAR-10 experiments (Table 5), the results for the different methods are within confidence intervals. Since the selective classification errors are very small, it is difficult to draw conclusions from such results. On CIFAR-10, SAT achieves 99+% accuracy at 80% coverage. In contrast, on

| # Classes | 30% Coverage | | 50% Coverage | | 70% Coverage | |
|---|---|---|---|---|---|---|
| | SAT | SAT+EM+SR | SAT | SAT+EM+SR | SAT | SAT+EM+SR |
| 175 | 0.69 ± 0.12 | **0.46 ± 0.05** | **1.27 ± 0.12** | **0.91 ± 0.16** | 3.03 ± 0.13 | **2.73 ± 0.07** |
| 150 | 0.44 ± 0.13 | **0.16 ± 0.02** | 0.81 ± 0.11 | **0.47 ± 0.05** | 2.23 ± 0.16 | **1.71 ± 0.15** |
| 125 | 0.44 ± 0.07 | **0.14 ± 0.09** | 0.93 ± 0.11 | **0.52 ± 0.07** | 2.32 ± 0.25 | **1.84 ± 0.14** |
| 100 | 0.71 ± 0.11 | **0.15 ± 0.06** | 1.11 ± 0.10 | **0.56 ± 0.06** | 2.65 ± 0.19 | **1.81 ± 0.08** |
| 75 | 0.50 ± 0.15 | **0.09 ± 0.00** | 1.01 ± 0.08 | **0.40 ± 0.03** | 2.60 ± 0.15 | **1.68 ± 0.27** |
| 50 | 0.76 ± 0.06 | **0.16 ± 0.05** | 1.44 ± 0.30 | **0.37 ± 0.10** | 2.86 ± 0.34 | **1.47 ± 0.12** |
| 25 | 0.53 ± 0.00 | **0.08 ± 0.11** | 0.64 ± 0.23 | **0.21 ± 0.08** | 1.79 ± 0.53 | **1.14 ± 0.25** |

Table 6: Comparison between the Selective classification error for Self-Adaptive Training (SAT) and SAT with Entropy Minimization (EM) and Softmax Response (SR) on ImagenetSubset.

Imagenet100, SAT achieves 95% at 80% coverage. The saturation of CIFAR-10 is further highlighted in previous works which show improvements on the dataset (Geifman & El-Yaniv, 2019; Ziyin et al., 2019; Huang et al., 2022) on the scale of $0.1 - 0.2\%$.

### 5.3.3 SCALABILITY WITH THE NUMBER OF CLASSES: IMAGENETSUBSET

To evaluate the scalability of the proposed methodology with respect to the number of classes, we evaluate our proposed method SAT+EM+SR with the previous state-of-the-art SAT on ImagenetSubset. In Table 6, we see once again that Self-Adaptive Training with our proposed entropy-regularised loss function and selecting according to Softmax Response outperforms the previous state-of-the-art (vanilla Self-Adaptive Training) by a very significant margin (up to $85\%$ relative improvement) across all sizes of datasets. Due to the space limitations, the results for the other coverages of Table 6 are included in the Appendix.

### 5.3.4 ENTROPY-MINIMIZATION ONLY, SOFTMAX RESPONSE SELECTION ONLY, OR BOTH?

In this experiment, we show that applying EM or SR alone provide gains. However, to achieve state-of-the-art results by a large margin, it is crucial to use the combination of both SR and EM. Table 4 shows that using only the entropy-minimization (SAT-EM) slightly improves the performance of SAT. However, SAT+EM+SR (SAT+EM in conjunction with SR selection mechanism) improves upon SAT+SR and SAT+EM significantly, achieving new state-of-the art results for selective classification.

## 6 CONCLUSION

In this work, we analysed the state-of-the-art Selective Classification methods and concluded that their strong performance is owed to learning a more generalisable classifier rather, yet their suggested selective solutions are suboptimal. Accordingly, we showed that selection mechanisms based on the classifier itself outperforms the state-of-the-art selection methods. These results suggest that future work in selective classification should explore selection mechanisms based on the classifier itself rather than following recent works which proposed architecture modifications. Moreover, we also highlighted a connection between selective classification and semi-supervised learning, which to our knowledge has not been explored before. We show that a common technique in semi-supervised learning, namely, entropy-minimization, greatly improves performance in selective classification, opening the door to further exploration of the relationship between these two fields.

From a practical perspective, we showed that selecting according to classification scores is the SOTA selection mechanism for comparison. Importantly, this method can be applied to an already deployed trained selective classification model and instantly improve performance at negligible cost. In addition, we showed a selective classifier trained with the entropy-regularised loss and with selection according to Softmax Response achieves new state-of-the-art results by a significant margin.

### REPRODUCIBILITY STATEMENT

In our experiments, we build on the official implementations of Self-Adaptive Training available at https://github.com/LayneH/SAT-selective-cls. Our code is available at

https://github.com/BorealisAI/towards-better-sel-cls. The experiments with Deep Gamblers (Link: https://github.com/Z-T-WANG/NIPS2019DeepGamblers) are run using the official implementaiton. Our Pytorch implementation of SelectiveNet follows the details in the original paper. The implementation details are available in Section 4. The hyperparameters are available in Section 5 and Appendix C.

## ACKNOWLEDGEMENTS

The authors acknowledge funding from the Quebec government.

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

# A  APPENDIX: ADDITIONAL BACKGROUND AND BROADER IMPACT

## A.1  DEEP GAMBLERS

Inspired by portfolio theory, Deep Gamblers proposes to train the model using the following loss function:

$$\mathcal{L} = -\frac{1}{m} \sum_{i=1}^{m} p(y_i|x) \log \left( p_\theta(y_i|x) + \frac{1}{o} p_\theta(C+1|x) \right),$$

where $m$ is the number of datapoints in the batch and $o$ is a hyperparameter controlling the impact of the abstain logit. Smaller values of $o$ encourages the model to abstain more often. However, $o \leq 1$ makes it ideal to abstain for all datapoints and $o > C$ makes it ideal to predict for all datapoints. As a result, $o$ is restricted to be between 1 and $C$. Note that the corresponding loss function with large values of $o$ is approximately equivalent to the Cross Entropy loss.

## A.2  BROADER IMPACT

The broader impact of this work depends on the application of the selective model. In terms of the societal impact, fairness in selection remains a concern as lowering the coverage can magnify the difference in recall between groups and increase unfairness (Jones et al., 2021; Lee et al., 2021).

The calibration step performed on the validation set assumes the validation and test data are sampled from the same distribution. Hence, in the case of out-of-distribution test data, a selective classifier calibrated to, for example, 70

When evaluating, Selective Classifiers may choose to predict samples with easier classes more than hard to predict classes. Thus, it would be undesirable in fairness applications that require equal coverage amongst the different classes.

# B  APPENDIX: ALTERNATIVE MOTIVATION

In the selective classification problem setting, the objective is to select $c_{\text{target}}$ proportion of samples for prediction according to the value outputted by a selection function, $\bar{g}(x)$. Since each datapoint $(x_i, y_i)$ is an i.i.d. sample, it is optimal to iteratively select from the dataset $\mathcal{D}$ the sample $x^*$ that maximizes the selection function, *i.e.*, $x^* \in \text{argmax}_{x \in \mathcal{D}} \bar{g}(x)$., until the target coverage $c_{\text{target}}$ proportion of the dataset is reached. In other words, to select $c_{\text{target}}$ proportion of samples (coverage = $c_{\text{target}}$), it is sufficient to define the criterion $\bar{g}$ and select a threshold $\tau$ such that exactly $c_{\text{target}}$ proportion of samples satisfy $\bar{g}(x) > \tau$.

## B.1  SELECTING VIA PREDICTIVE ENTROPY

At test time, given a dataset of datapoints $\mathcal{D}$, if the labels were available, the optimal criterion to select the datapoint $x \in \mathcal{D}$ that minimise the loss function would be according to:

$$\text{argmin}_{x \in \mathcal{D}} \, CE \left( p(\cdot|x), p_\theta(\cdot|x) \right).$$

However, at test time, the labels are unavailable. Instead, we can use the model's belief over what the label is, *i.e.*, the learned approximation $p_\theta(\cdot|x) \approx p(\cdot|x)$. We know $CE(p_\theta(\cdot|x), p_\theta(\cdot|x)) = H(p_\theta(\cdot|x))$ where $H$ is the entropy function. As such, we can select samples according to

$$\text{argmin}_{x \in \mathcal{D}} \, CE \left( p(\cdot|x), p_\theta(\cdot|x) \right) \approx \text{argmin}_{x \in \mathcal{D}} \, H \left( p_\theta(\cdot|x) \right).$$

In other words, entropy is an approximation for the unknown loss function. Accordingly, with respect to the discussed selection framework (Section 3.1), the samples with the largest negative entropy value, *i.e.*, $\bar{g}(x) = -H(p_\theta(\cdot|x))$ are best nominees for selection.

In Figure 1a, we show the distribution of entropy for a trained vanilla classifier, empirically showing entropy to be strongly inversely correlated with the model's ability to correctly predict the labels. As a result, entropy is a good selection mechanism. We include results on CIFAR-10 and Imagenet100 for a vanilla classifier in Table 8 and Table 9.

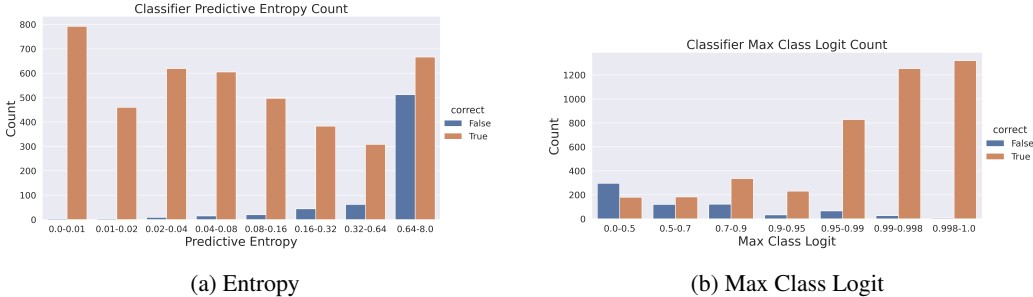

(a) Entropy                (b) Max Class Logit

Figure 1: A histogram of the number of datapoints according to a vanilla classifier trained on Imagenet100. The orange bar indicates the samples for which the model correctly predicts the class of the sample. The blue bar represents the samples for which the model incorrectly predicted the class. In the case of entropy, a lower value correponds to higher model confidence. In contrast, in the case of max class logit, a higher value correponds to higher model confidence.

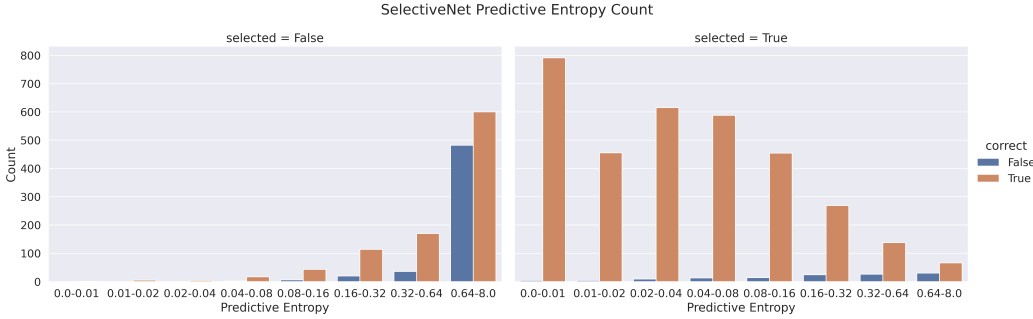

Figure 2: Entropy Comparison. SelectiveNet trained on Imagenet100 for a target coverage of $0.8$ and evaluated on a coverage of $0.8$. In the case of entropy, a lower value correponds to higher model confidence. The histogram represents the counts of samples that were incorrectly predicted by the model. The left image indicates datapoints that were not selected by the selection head, *i.e.*, datapoints with low selection value $h(x) < \tau$. The right image indicates datapoints that were selected by the selection head, *i.e.*, $h(x) \geq \tau$.

## B.2 SELECTING VIA MAXIMUM PREDICTIVE CLASS LOGIT (SOFTMAX RESPONSE)

Given a model with well-calibrated confidences (Guo et al., 2017; Minderer et al., 2021), an interpretation of $p_\theta(u|x)$ is a probability estimate of the true correctness likelihood, i.e., $p_\theta(u|x)$ is the likelihood that $u$ is the correct label of $x$. Let $y_i$ be the correct label for $x_i$. For example, given 100 samples $\{x_1, \ldots, x_{100}\}$ with $p_\theta(u|x_i) = 0.8$, we would expect approximately $80\%$ of the samples to have $u$ as its label. As a result, $p_\theta(y_i|x_i)$ is the model's probability estimate that the correct label is $y_i$. In classification, the probability that the calibrated model predicts a datapoint $x$ correctly is equivalent to the value of the max class logit, *i.e.*, $\max_{u \in \{1, \ldots C\}} p_\theta(u|x)$. Logically, the sample $x_i$ that should be selected for clasification is the sample the model's most likely to predict the sample correctly, *i.e.*, $i = \arg\max_j \left( \max_{u \in \{1, \ldots C\}} p_\theta(u|x_j) \right)$. This selection is equivalent to selecting according to the following soft selection function $\bar{g}(x) = \max_{u \in \{1, \ldots C\}} p_\theta(u|x)$. Simply put, this is equivalent to selecting according to the maximum predictive class logit (aka Softmax Response (Geifman & El-Yaniv, 2017)).

In practice, neural network models are not guaranteed to have well-calibrated confidences. In Selective Classification, however, we threshold according to $\tau$ and select samples above the threshold $\tau$ for classification, so we do not use the exact values of the confidence (max class logit). As a result, we do not need the model to necessarily have well-calibrated confidences. Instead, it suffices if samples with higher confidences (max class logit) have a higher likelihood of being correct. In Figure 1b, we show the distribution of max class logit for a trained vanilla classifier, empirically showing larger max class logit to be strongly correlated with model's ability to correctly predict the label.

As a result, max class logit is a good selection mechanism. We include results on CIFAR-10 and Imagenet100 for a vanilla classifier in Table 8 and Table 9.

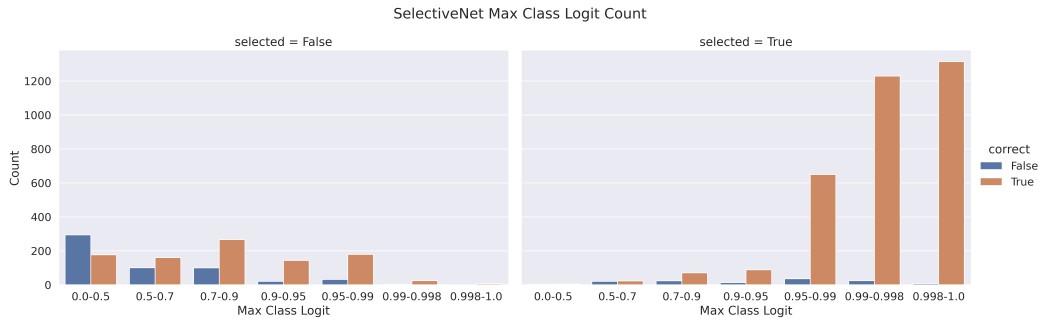

Figure 3: Max Class Logit Comparison. SelectiveNet trained on Imagenet100 for a target coverage of $0.8$ and evaluated on a coverage of $0.8$. In the case of max class logit, a higher value correponds to higher model confidence. The histogram represents the counts of samples that were incorrectly predicted by the model. The left image indicates datapoints that were not selected by the selection head, *i.e.*, datapoints with low selection value $h(x) < \tau$. The right image indicates datapoints that were selected by the selection head, *i.e.*, $h(x) \geq \tau$

### B.3    RECIPE FOR BETTER SELECTIVE CLASSIFICATION

In this section, we further illustrate how SelectiveNet's original selection mechanism is suboptimal. The optimisation of SelectiveNet's selective loss $\mathcal{L}_{selective}$ (See Section 3.2.1) aims to learn a selection head (soft selection model) $\bar{g}$ that outputs a low selection value for inputs with large cross-entropy loss and high selection value for inputs with low cross-entropy loss. At test time, good performance of SelectiveNet depends on the generalisation of both the prediction and selection heads. However, learned models can at times fail to generalise. In Figure 2 and Figure 3, we show the distribution of entropy and max class logit for selected and not-selected samples according to a SelectiveNet model. In the plots, we see SelectiveNet's original selection mechanism selects several samples with large entropy and low max class logit. In Table 2, we see that the selection mechanisms based on entropy and max class logit outperforms the original selection mechanism. This comparison further supports our argument that the selection mechanism should be rooted in the objective function instead of a separately calculated score.

## C    APPENDIX: ADDITIONAL EXPERIMENTAL DETAILS

### C.1    HYPERPARAMETERS

Following (Geifman & El-Yaniv, 2019), SelectiveNet was trained with a target coverage rate and evaluated on the same coverage rate. As a result, there are different models for each experimental coverage rate. In contrast, target coverage does not play a role in the optimization process of Deep Gamblers and Self-Adaptive Training, hence, the results for different experimental coverages are computed with the same models.

All CIFAR-10 experiments were performed with 5 seeds. All Imagenet-related experiments were performed with 3 seeds. For hyperparameter tuning, we split Imagenet100's training data into $80\%$ training data and $20\%$ validation data evenly across the different classes. We tested the following values for the entropy minimization coefficient $\beta \in \{0.1, 0.01, 0.001, 0.0001\}$. For the final evaluation, we trained the model on the entire training data.

Self-Adaptive Training models are trained using SGD with an initial learning rate of $0.1$ and a momentum of $0.9$.

**Food101/Imagenet100/ImagenetSubset.** The models were trained for 500 epochs with a mini-batch size of 128. The learning rate was reduced by 0.5 every 25 epochs. The entropy-minimization term was $\beta = 0.01$.

**CIFAR-10.** The models were trained for 300 epochs with a mini-batch size of 64. The learning rate was reduced by 0.5 every 25 epochs. The entropy-minimization term was $\beta = 0.001$.

**StanfordCars.** The models were trained for 300 epochs with a mini-batch size of 64. The learning rate was reduced by 0.5 every 25 epochs. The entropy-minimization term was $\beta = 0.01$.

**Imagenet.** The models were trained for 150 epochs with a mini-batch size of 256. The learning rate was reduced by 0.5 every 10 epochs. The entropy-minimization term was $\beta = 0.001$.

## C.2 COMPUTE

The experiments were primarily run on a GTX 1080 Ti. The CIFAR10 experiments took  1.5 hours for Self-Adaptive Training and Deep Gamblers. SelectiveNet experiments took  3 hours each. The Imagenet100 experiments took  2 days for Self-Adaptive Training and Deep Gamblers. SelectiveNet experiments took  2.75 days each. The ImagenetSubset experiments took 0.5-4.5 days each for Self-Adaptive Training and Deep Gamblers, depending on the number of classes. SelectiveNet experiments took 0.75-5.5 days each, depending on the number of classes.

## C.3 IMAGENETSUBSET: CLASSES

ImagenetSubset comprises of multiple datasets ranging from 25 to 175 classes in increments of 25, *i.e.*, $\{\mathcal{D}_{25}, \mathcal{D}_{50}, \mathcal{D}_{75}, \mathcal{D}_{100}, \mathcal{D}_{125}, \mathcal{D}_{150}, \mathcal{D}_{175}\}$. Let $C_{25}, C_{50}, \ldots, C_{175}$ represent the classes of the respective datasets. The classes for ImagenetSubset are uniform randomly sampled from the classes of Imagenet such that the classes of the smaller datasets are subsets of the classes of the larger datasets, *i.e.* $\mathcal{D}_{25} \subset \mathcal{D}_{50} \subset \mathcal{D}_{75} \subset \cdots \subset \mathcal{D}_{175}$ and $C_{25} \subset C_{50} \ldots C_{175}$. The list of Imagenet classes in each dataset is included below for reproducibility.

### C.3.1 $C_{25}$

n03133878 n03983396 n03995372 n03776460 n02730930 n03814639 n03666591 n03110669
n04442312 n02017213 n04265275 n01774750 n03709823 n09256479 n07715103 n04560804
n02120505 n04522168 n04074963 n02268443 n03291819 n02091467 n02486261 n03180011
n02100236

### C.3.2 $C_{50} - C_{25}$

n02106662 n01871265 n12057211 n04579432 n07734744 n02408429 n02025239 n03649909
n03041632 n02484975 n02097209 n03854065 n03476684 n04579145 n01739381 n02319095
n01843383 n02229544 n09288635 n02138441 n02119022 n07583066 n03534580 n02817516
n04356056

### C.3.3 $C_{75} - C_{50}$

n03424325 n04507155 n02112350 n03450230 n01616318 n01641577 n03630383 n01530575
n02102973 n04310018 n02134084 n01729322 n03250847 n02099849 n03544143 n03871628
n03777754 n04465501 n01770081 n03255030 n01910747 n03016953 n03485407 n03998194
n02129604

### C.3.4 $C_{100} - C_{75}$

n02128757 n03763968 n01677366 n03483316 n02177972 n03814906 n01753488 n02116738
n01755581 n02264363 n03290653 n13133613 n03929660 n04040759 n02317335 n02494079
n02865351 n03134739 n02102177 n04192698 n02814533 n04090263 n01818515 n01748264
n04328186

### C.3.5 $C_{125} - C_{100}$

n03930313 n02422106 n07714571 n02111277 n03706229 n03729826 n03344393 n07831146
n02090379 n06596364 n03187595 n04317175 n11939491 n04277352 n01807496 n02804610

| Coverage | Self-Adaptive Training | Deep Gamblers | SelectiveNet | MC-Dropout |
|---|---|---|---|---|
| 100 | **5.91 ± 0.04** | 6.08 ± 0.00 | 6.47 ± 0.22 | 6.79 ± 0.03 |
| 95 | **3.73 ± 0.13** | 3.71 ± 0.00 | 4.07 ± 0.12 | 4.58 ± 0.05 |
| 90 | **2.18 ± 0.11** | 2.27 ± 0.00 | 2.49 ± 0.13 | 2.92 ± 0.01 |
| 85 | **1.26 ± 0.09** | 1.29 ± 0.00 | 1.42 ± 0.08 | 1.82 ± 0.09 |
| 80 | **0.69 ± 0.04** | 0.81 ± 0.00 | 0.86 ± 0.05 | 1.08 ± 0.05 |
| 75 | **0.37 ± 0.01** | 0.44 ± 0.00 | 0.53 ± 0.06 | 0.66 ± 0.05 |
| 70 | **0.27 ± 0.02** | 0.30 ± 0.00 | 0.42 ± 0.04 | 0.43 ± 0.05 |

Table 7: Comparison of existing Selective Classification baselines with MC-Dropout. The results of MC-Dropout are originally from Geifman & El-Yaniv (2017). For a given coverage, the **bolded** result indicate the lowest selective risk (i.e. best result) and underlined result indicate the second lowest selective risk.

n02093991 n09428293 n03207941 n02132136 n04548280 n02793495 n03924679 n02112137 n02107312

### C.3.6 $C_{150} - C_{125}$

n03376595 n03467068 n02837789 n04467665 n04243546 n03530642 n04398044 n02113624 n13044778 n03188531 n01729977 n01980166 n02101388 n01629819 n01773157 n01689811 n02109525 n03938244 n02123045 n04548362 n04612504 n04264628 n02108551 n04311174 n02276258

### C.3.7 $C_{175} - C_{150}$

n03724870 n02087046 n09421951 n02799071 n07717410 n02906734 n02206856 n03877472 n01740131 n04523525 n03496892 n04116512 n03743016 n03759954 n04462240 n03788195 n02137549 n03866082 n02233338 n02219486 n02445715 n02974003 n01924916 n12620546 n02992211

## D  APPENDIX: ADDITIONAL RESULTS

Briefly summarised, the additional interesting results found are as follows: (1) In low coverage settings, selecting based on Softmax Response and Entropy on a vanilla classifier trained via the cross entropy loss outperform both SelectiveNet and Deep Gamblers. (2) SelectiveNet outperforms Deep Gamblers on moderate coverages ($60\%$, $50\%$, and $40\%$) which is not in par with the previously reported results. We attribute the interesting results to our work being the first to evaluate these methods on large datasets at a wide range of coverages. Since previous works have mainly focused on toy datasets and high coverages ($70 + \%$), they failed to capture these patterns. The main takeaway of these results, however, is that, across all the reported methods, selecting via Softmax Response is best.

### D.1  ANALYSIS OF SELECTED IMAGES: SELECTIVENET

We include in Figure 4 and Figure 5 examples of images where the selection head of SelectiveNet fails to generalise.

### D.2  SELECTION MECHANISMS: MC-DROPOUT

In Table 7, we see that MC-Dropout performs worse than the existing state-of-the-art methods for Selective Classification.

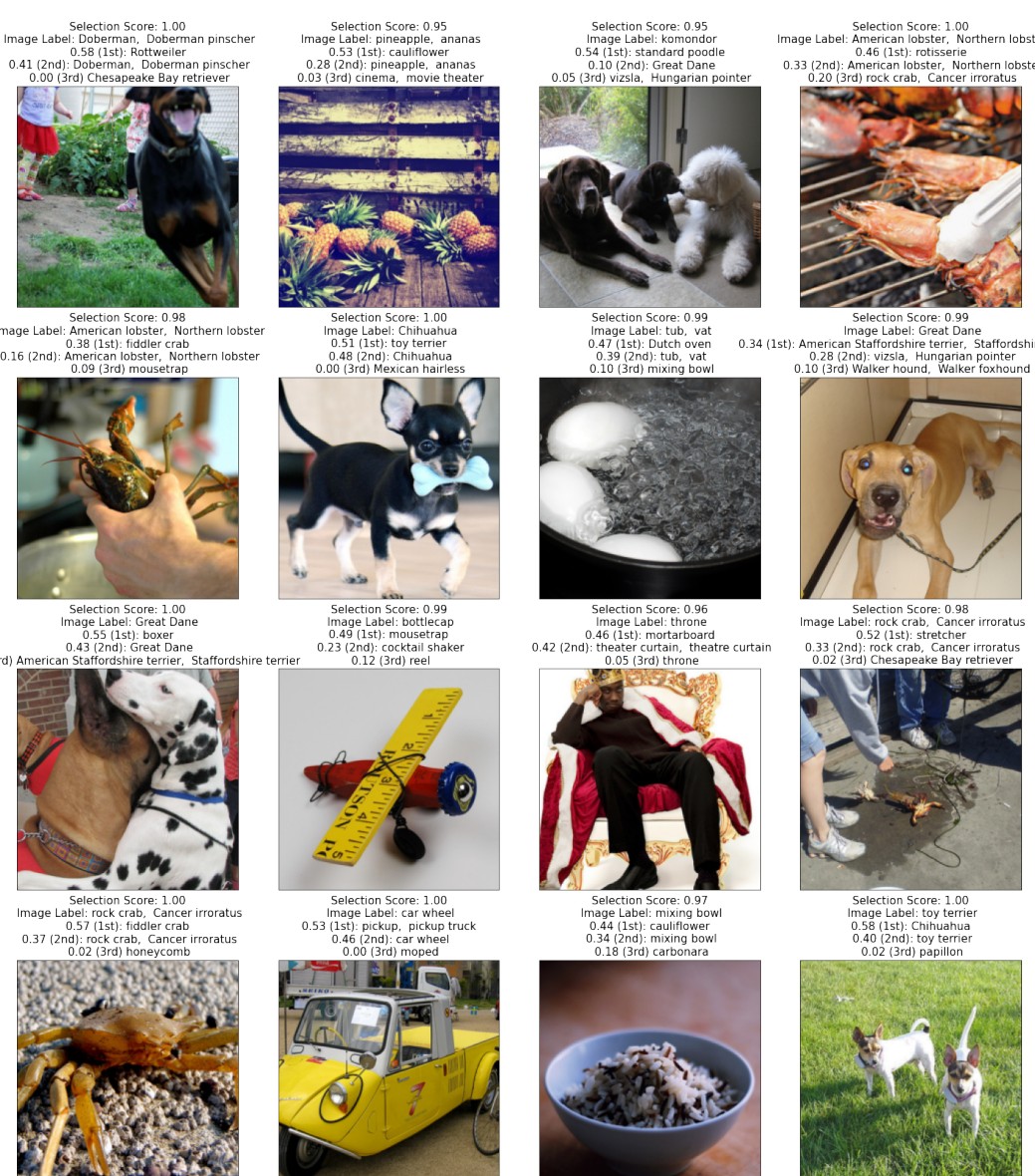

Figure 4: SelectiveNet (at $80\%$ coverage) on Imagenet100: Incorrect and unconfident predictions according to its classifier but selected images according to its Selective Head. The selection score threshold for selecting images for prediction is $0.93$. Included are the selection score according to the Selection Head, the image's label, and the top-3 predicted classes according to the classifier and their respective classifier scores.

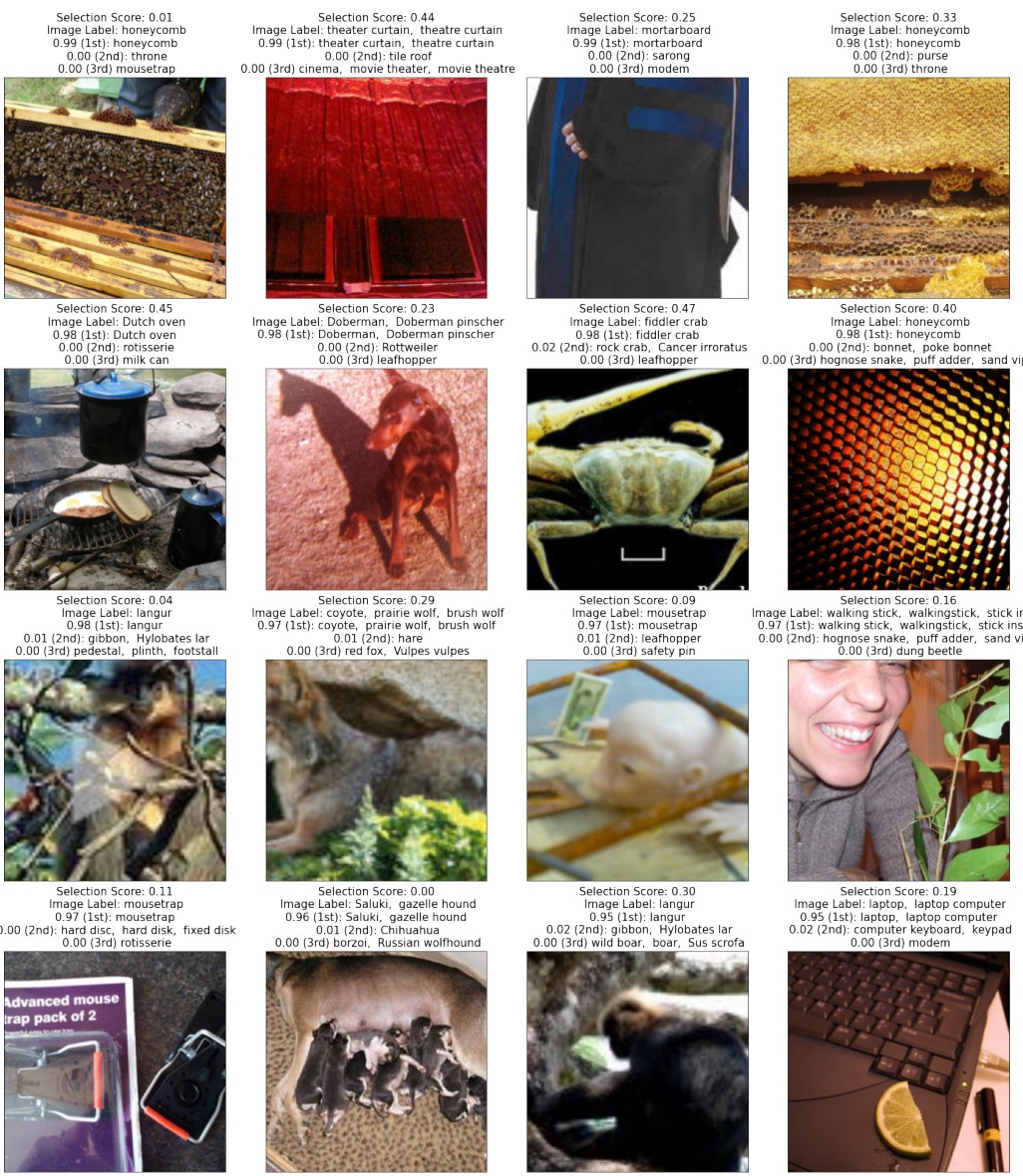

Figure 5: SelectiveNet (at $80\%$ coverage) on Imagenet100: Correct and confident predictions according to its classifier but rejected images according to its Selective Head.

| Dataset | Coverage | Vanilla Classifier | |
| | | Entropy | Softmax Response |
| --- | --- | --- | --- |
| CIFAR-10 | 100 | **6.61 ± 0.25** | **6.61 ± 0.25** |
| | 95 | **4.30 ± 0.19** | **4.35 ± 0.17** |
| | 90 | **2.63 ± 0.12** | **2.63 ± 0.11** |
| | 85 | **1.62 ± 0.10** | **1.63 ± 0.12** |
| | 80 | **1.01 ± 0.13** | **0.99 ± 0.10** |
| | 75 | **0.72 ± 0.08** | **0.72 ± 0.08** |
| | 70 | **0.57 ± 0.08** | **0.55 ± 0.07** |

Table 8: Comparison of selection based on Entropy and Softmax Response for a vanilla classifier trained with cross-entropy loss on CIFAR-10.

| Dataset | Coverage | Vanilla Classifier | |
| | | Entropy | Softmax Response |
| --- | --- | --- | --- |
| Imagenet100 | 100 | **14.32 ± 0.14** | **14.32 ± 0.14** |
| | 90 | **9.14 ± 0.05** | **8.96 ± 0.13** |
| | 80 | 5.34 ± 0.12 | **4.99 ± 0.05** |
| | 70 | **3.04 ± 0.14** | **2.83 ± 0.12** |
| | 60 | **1.80 ± 0.14** | **1.70 ± 0.19** |
| | 50 | **1.22 ± 0.31** | **1.08 ± 0.28** |
| | 40 | **0.82 ± 0.32** | **0.77 ± 0.39** |
| | 30 | **0.63 ± 0.33** | **0.60 ± 0.28** |
| | 20 | **0.60 ± 0.28** | **0.60 ± 0.28** |
| | 10 | **0.30 ± 0.14** | **0.20 ± 0.28** |

Table 9: Comparison of selection based on Entropy and Softmax Response for a vanilla classifier trained with cross-entropy loss on Imagenet100.

### D.3 SELECTION MECHANISMS: VANILLA CLASSIFIER

#### D.3.1 CIFAR-10

In Table 8, the difference in performance between selecting according to entropy and selecting according to Softmax Response is not significant. We attribute this marginal difference to the saturatedness of the CIFAR-10 dataset.

#### D.3.2 IMAGENET100

In Table 9, we see that selecting according to Softmax Response clearly outperforms selecting according to entropy. We see that Softmax Response learns a less generalizeable clasifier (See performance on $100\%$ coverage) than Self-Adaptive Training, Deep Gamblers, and SelectiveNet. However, interestingly, we found that Softmax Response outperforms both Deep Gamblers and SelectiveNet on low coverages ($10\%$, $20\%$, $30\%$). Previous works failed to capture this pattern due to lack of evaluation on larger datasets and lower coverages.

### D.4 SELECTION MECHANISMS: DEEP GAMBLERS

**CIFAR-10.** In these results (Table 10), we see that the difference in performance between the various selection mechanisms is marginal. Due to the marginal difference between errors, it is difficult to make conclusions from these results.

**Imagenet100.** In Table 10, we see that selecting according to Softmax Response and Entropy clearly outperforms the original selection mechanism.

**ImagenetSubset.** In Table 11, similar to Imagenet100, we see a clear substantial improvement when using Softmax Response as the selection mechanism instead of the original selection mechanism. Furthermore, we see that Entropy also outperforms the original selection mechanism.

| Dataset | Coverage | Deep Gamblers | | |
|---------|----------|------|--------------|---------|
|         |          | DG | DG + Entropy | DG + SR |
| CIFAR-10 | 100 | **6.08 ± 0.00** | **6.08 ± 0.00** | **6.08 ± 0.00** |
|         | 95 | **3.71 ± 0.00** | 3.79 ± 0.00 | 3.81 ± 0.00 |
|         | 90 | 2.27 ± 0.00 | **2.14 ± 0.00** | 2.16 ± 0.00 |
|         | 85 | **1.29 ± 0.00** | 1.31 ± 0.00 | 1.35 ± 0.00 |
|         | 80 | **0.81 ± 0.00** | 0.84 ± 0.00 | 0.85 ± 0.00 |
|         | 75 | **0.44 ± 0.00** | 0.57 ± 0.00 | 0.56 ± 0.00 |
|         | 70 | **0.30 ± 0.00** | 0.41 ± 0.00 | 0.43 ± 0.00 |
| Imagenet100 | 100 | **13.49 ± 0.52** | **13.49 ± 0.52** | **13.49 ± 0.52** |
|         | 90 | 8.42 ± 0.44 | 8.25 ± 0.43 | **8.11 ± 0.48** |
|         | 80 | 5.21 ± 0.32 | 4.76 ± 0.37 | **4.52 ± 0.38** |
|         | 70 | 3.30 ± 0.40 | 2.70 ± 0.21 | **2.58 ± 0.21** |
|         | 60 | 2.14 ± 0.37 | 1.86 ± 0.32 | **1.71 ± 0.32** |
|         | 50 | 1.55 ± 0.27 | 1.35 ± 0.25 | **1.31 ± 0.22** |
|         | 40 | 1.23 ± 0.38 | 1.20 ± 0.11 | **1.07 ± 0.19** |
|         | 30 | 1.09 ± 0.31 | 1.00 ± 0.19 | **0.96 ± 0.21** |
|         | 20 | 1.03 ± 0.31 | 0.97 ± 0.21 | **0.90 ± 0.22** |
|         | 10 | 0.80 ± 0.28 | 0.73 ± 0.25 | **0.53 ± 0.25** |

Table 10: Deep Gamblers Results on CIFAR-10 and Imagenet100. Comparison of Selection Mechanism Results.

| Dataset | # of Classes | Deep Gamblers | | |
|---------|--------------|------|--------------|---------|
|         |              | DG | DG + Entropy | DG + SR |
| ImagenetSubset | 175 | **3.77 ± 0.10** | **3.75 ± 0.14** | **3.62 ± 0.11** |
|         | 150 | **2.62 ± 0.03** | **2.65 ± 0.26** | **2.54 ± 0.24** |
|         | 125 | 2.58 ± 0.25 | **2.40 ± 0.19** | **2.22 ± 0.17** |
|         | 100 | 2.57 ± 0.04 | 2.30 ± 0.01 | **2.20 ± 0.04** |
|         | 75 | 2.60 ± 0.20 | 2.29 ± 0.00 | **2.22 ± 0.05** |
|         | 50 | 2.63 ± 0.12 | **2.15 ± 0.05** | **2.08 ± 0.07** |
|         | 25 | 1.60 ± 0.28 | **1.22 ± 0.30** | **1.30 ± 0.19** |

Table 11: Deep Gamblers Results on ImagenetSubset (70% coverage) with various selective mechanisms.

## D.5  SELECTION MECHANISMS: SELF-ADAPTIVE TRAINING

**ImagenetSubset.** In addition to the Imagenet100 experiments, we also evaluate Self-Adaptive Training trained with the proposed entropy-regularised loss function on ImagenetSubset.

In Figure 6 (and Table 12 and Table 13), we see that training with the entropy-regularised loss function improves the scalability of Self-Adaptive Training when selecting according to Softmax Response.

In Figure 6, the results for SelectiveNet, Deep Gamblers, and Self-Adaptive Training on 70% coverage. Consistent with previous experiments, we see that both the selection mechanisms based on the classifier itself (predictive entropy and Softmax Response) significantly outperform the original selection mechanisms of the proposed methods. These results further support our conclusion that (1) the strong performance of these methods were due to them learning a more generalizable model and (2) the selection mechanism should stem from the classifier itself rather than a separate head/logit. Similarly, we see that Softmax Response is the state-of-the-art selection mechanism. In the experiments, we see that SelectiveNet struggles to scale to harder tasks. Accordingly, the achieved improvement in selective accuracy with Softmax Response (SR) increases as the number of classes increase. This suggests that the proposed selection mechanism is more beneficial for SelectiveNet as the difficulty of the task increases, *i.e.*, improves scalability.

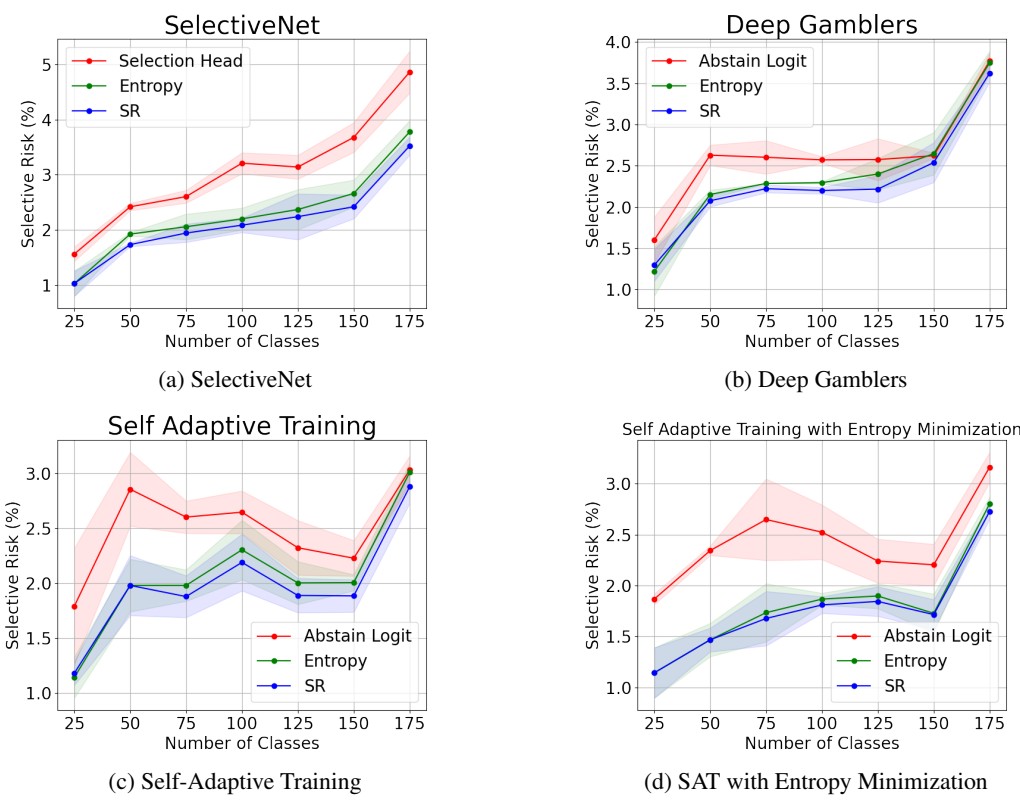

(a) SelectiveNet

(b) Deep Gamblers

(c) Self-Adaptive Training

(d) SAT with Entropy Minimization

Figure 6: ImagenetSubset at 70% coverage. (a), (b), and (c) Various Selective Models (d) Comparison of Self-Adaptive Training trained with the proposed entropy-regularised loss and without. The loss function improves the scalability of Self-Adaptive Training, particularly when using Softmax Response as the selection mechanism.

| # Classes | SAT | | SAT + Entropy | | SAT + Softmax Response | |
|---|---|---|---|---|---|---|
| | SAT | SAT + EM | SAT + E | SAT+EM+E | SAT + SR | SAT+EM+SR |
| 175 | 3.03 ± 0.13 | 3.16 ± 0.15 | 3.01 ± 0.09 | 2.80 ± 0.07 | 2.88 ± 0.15 | **2.73 ± 0.07** |
| 150 | 2.23 ± 0.16 | 2.20 ± 0.20 | 2.01 ± 0.07 | 1.73 ± 0.19 | 1.89 ± 0.15 | **1.71 ± 0.15** |
| 125 | 2.32 ± 0.25 | 2.24 ± 0.22 | 2.00 ± 0.19 | 1.90 ± 0.12 | 1.89 ± 0.16 | **1.84 ± 0.14** |
| 100 | 2.65 ± 0.19 | 2.52 ± 0.27 | 2.30 ± 0.27 | 1.87 ± 0.06 | 2.19 ± 0.26 | **1.81 ± 0.08** |
| 75 | 2.60 ± 0.15 | 2.65 ± 0.40 | 1.98 ± 0.14 | 1.73 ± 0.29 | 1.88 ± 0.19 | **1.68 ± 0.27** |
| 50 | 2.86 ± 0.34 | 2.34 ± 0.05 | 1.98 ± 0.24 | **1.47 ± 0.16** | 1.98 ± 0.27 | **1.47 ± 0.12** |
| 25 | 1.79 ± 0.53 | 1.87 ± 0.05 | **1.14 ± 0.19** | 1.14 ± 0.25 | 1.18 ± 0.11 | **1.14 ± 0.25** |

Table 12: Self Adaptive Training with Entropy Minimization Loss Function on ImagenetSubset at 70% coverage. We see that SAT+EM+SR performs the best and outperforms SAT by a statistically significant margin.

### D.6 ABLATION: VARYING ARCHITECTURE

In these experiments, we show generalizability across architectures of our proposed entropy-minimization and softmax response methodology. In Tables 14, 15, and 16, we see that applying the entropy-minimization and Softmax Response methodology improves upon the state-of-the-art method's performance significantly.

### D.7 RISK-COVERAGE PLOTS

Figure 7 shows the risk coverage plots for Imagenet100, Food101, and StanfordCars results.

### D.8 LEARNING CURVES PLOTS

Figure 8 shows that SAT and SAT+EM models have converged on StanfordCars.

|  |  |  | Self-Adaptive Training | |
|---|---|---|---|---|
| Dataset | Coverage | # of Classes | SAT | SAT + EM + SR |
| ImagenetSubset | 30 | 175 | 0.69 ± 0.12 | **0.46 ± 0.05** |
|  |  | 150 | 0.44 ± 0.13 | **0.16 ± 0.02** |
|  |  | 125 | 0.44 ± 0.07 | **0.14 ± 0.09** |
|  |  | 100 | 0.71 ± 0.11 | **0.15 ± 0.06** |
|  |  | 75 | 0.50 ± 0.15 | **0.09 ± 0.00** |
|  |  | 50 | 0.76 ± 0.06 | **0.16 ± 0.05** |
|  |  | 25 | 0.53 ± 0.00 | **0.08 ± 0.11** |
|  | 40 | 175 | 0.94 ± 0.06 | **0.59 ± 0.14** |
|  |  | 150 | 0.64 ± 0.03 | **0.34 ± 0.06** |
|  |  | 125 | 0.76 ± 0.06 | **0.25 ± 0.04** |
|  |  | 100 | 0.90 ± 0.15 | **0.30 ± 0.00** |
|  |  | 75 | 0.84 ± 0.14 | **0.23 ± 0.03** |
|  |  | 50 | 1.17 ± 0.39 | **0.27 ± 0.13** |
|  |  | 25 | 0.67 ± 0.25 | **0.07 ± 0.09** |
|  | 50 | 175 | 1.27 ± 0.12 | **0.91 ± 0.16** |
|  |  | 150 | 0.81 ± 0.11 | **0.47 ± 0.05** |
|  |  | 125 | 0.93 ± 0.11 | **0.52 ± 0.07** |
|  |  | 100 | 1.11 ± 0.10 | **0.56 ± 0.06** |
|  |  | 75 | 1.01 ± 0.08 | **0.40 ± 0.03** |
|  |  | 50 | 1.44 ± 0.30 | **0.37 ± 0.10** |
|  |  | 25 | 0.64 ± 0.23 | **0.21 ± 0.08** |
|  | 60 | 175 | 1.77 ± 0.12 | **1.44 ± 0.20** |
|  |  | 150 | 1.21 ± 0.10 | **0.87 ± 0.04** |
|  |  | 125 | 1.34 ± 0.17 | **0.95 ± 0.01** |
|  |  | 100 | 1.67 ± 0.07 | **0.93 ± 0.03** |
|  |  | 75 | 1.51 ± 0.18 | **0.78 ± 0.02** |
|  |  | 50 | 1.78 ± 0.16 | **0.69 ± 0.08** |
|  |  | 25 | 0.93 ± 0.19 | **0.49 ± 0.17** |
|  | 70 | 175 | 3.03 ± 0.13 | **2.73 ± 0.07** |
|  |  | 150 | 2.23 ± 0.16 | **1.71 ± 0.15** |
|  |  | 125 | 2.32 ± 0.25 | **1.84 ± 0.14** |
|  |  | 100 | 2.65 ± 0.19 | **1.81 ± 0.08** |
|  |  | 75 | 2.60 ± 0.15 | **1.68 ± 0.27** |
|  |  | 50 | 2.86 ± 0.34 | **1.47 ± 0.12** |
|  |  | 25 | 1.79 ± 0.53 | **1.14 ± 0.25** |
|  | 80 | 175 | 5.85 ± 0.13 | **5.37 ± 0.15** |
|  |  | 150 | 4.46 ± 0.05 | **3.88 ± 0.19** |
|  |  | 125 | 4.78 ± 0.26 | **3.94 ± 0.34** |
|  |  | 100 | 4.94 ± 0.41 | **3.96 ± 0.02** |
|  |  | 75 | 4.91 ± 0.28 | **3.78 ± 0.35** |
|  |  | 50 | 5.05 ± 0.14 | **3.35 ± 0.36** |
|  |  | 25 | 4.13 ± 0.34 | **2.80 ± 0.16** |
|  | 90 | 175 | **10.14 ± 0.32** | 9.69 ± 0.17 |
|  |  | 150 | **8.30 ± 0.20** | 8.08 ± 0.16 |
|  |  | 125 | 8.87 ± 0.04 | **8.18 ± 0.59** |
|  |  | 100 | **8.90 ± 0.54** | 8.18 ± 0.23 |
|  |  | 75 | **8.57 ± 0.44** | 7.78 ± 0.43 |
|  |  | 50 | 8.79 ± 0.17 | **6.96 ± 0.64** |
|  |  | 25 | 7.79 ± 0.43 | **6.84 ± 0.19** |

Table 13: ImagenetSubset Results

|  | **ResNet34** | | |
|---|---|---|---|
| Coverage | SAT | SAT+SR | SAT+EM+SR |
| 100 | 37.68 ± 1.11 | 37.68 ± 1.11 | **32.49 ± 2.33** |
| 90 | 32.34 ± 1.19 | 32.04 ± 1.18 | **26.60 ± 2.39** |
| 80 | 26.86 ± 1.15 | 26.39 ± 1.13 | **20.87 ± 2.33** |
| 70 | 21.34 ± 1.20 | 20.70 ± 1.23 | **15.84 ± 1.98** |
| 60 | 16.21 ± 1.10 | 14.92 ± 1.03 | **11.09 ± 1.50** |
| 50 | 11.59 ± 0.74 | 10.25 ± 0.97 | **7.00 ± 1.13** |
| 40 | 7.76 ± 0.43 | 6.32 ± 0.69 | **4.00 ± 0.87** |
| 30 | 4.56 ± 0.35 | 3.54 ± 0.36 | **2.20 ± 0.44** |
| 20 | 2.42 ± 0.36 | 1.93 ± 0.09 | **1.17 ± 0.28** |
| 10 | 1.49 ± 0.00 | **1.20 ± 0.21** | **0.80 ± 0.22** |

Table 14: ResNet34: StanfordCars results

|  | **RegNetX** | | |
|---|---|---|---|
| Coverage | SAT | SAT+SR | SAT+EM+SR |
| 100 | **31.78 ± 2.44** | **31.78 ± 2.44** | 27.75 ± 1.81 |
| 90 | 26.35 ± 2.43 | **25.68 ± 2.44** | 21.72 ± 1.90 |
| 80 | 21.20 ± 2.40 | **20.07 ± 2.54** | 16.21 ± 1.79 |
| 70 | 16.45 ± 2.14 | **14.77 ± 2.23** | 11.22 ± 1.54 |
| 60 | 12.13 ± 1.64 | **10.07 ± 1.58** | 7.39 ± 1.21 |
| 50 | 8.60 ± 1.27 | **6.43 ± 1.46** | 4.55 ± 0.96 |
| 40 | 5.94 ± 1.06 | **4.04 ± 0.88** | 2.88 ± 0.61 |
| 30 | 3.99 ± 0.60 | **2.47 ± 0.44** | 1.74 ± 0.34 |
| 20 | 2.55 ± 0.33 | 1.55 ± 0.00 | **1.10 ± 0.34** |
| 10 | 1.66 ± 0.26 | **0.91 ± 0.15** | **0.70 ± 0.26** |

Table 15: RegNetX: StanfordCars results

|  | **ShuffleNet** | | |
|---|---|---|---|
| Coverage | SAT | SAT+SR | SAT+EM+SR |
| 100 | **34.10 ± 0.73** | **34.10 ± 0.73** | 32.90 ± 1.29 |
| 90 | **28.61 ± 0.72** | 28.27 ± 0.80 | 26.94 ± 1.33 |
| 80 | 23.16 ± 0.47 | **22.72 ± 0.63** | 21.13 ± 1.40 |
| 70 | 17.94 ± 0.27 | **17.14 ± 0.46** | 15.70 ± 1.42 |
| 60 | 13.00 ± 0.24 | **12.10 ± 0.46** | 10.89 ± 1.19 |
| 50 | 9.23 ± 0.10 | **7.68 ± 0.10** | 7.11 ± 0.87 |
| 40 | 6.31 ± 0.22 | **4.77 ± 0.24** | 4.49 ± 0.51 |
| 30 | 3.81 ± 0.39 | **2.97 ± 0.25** | 2.83 ± 0.28 |
| 20 | 2.07 ± 0.34 | **1.70 ± 0.30** | 1.43 ± 0.05 |
| 10 | **1.08 ± 0.26** | **0.99 ± 0.18** | 0.66 ± 0.31 |

Table 16: ShuffleNet: StanfordCars results

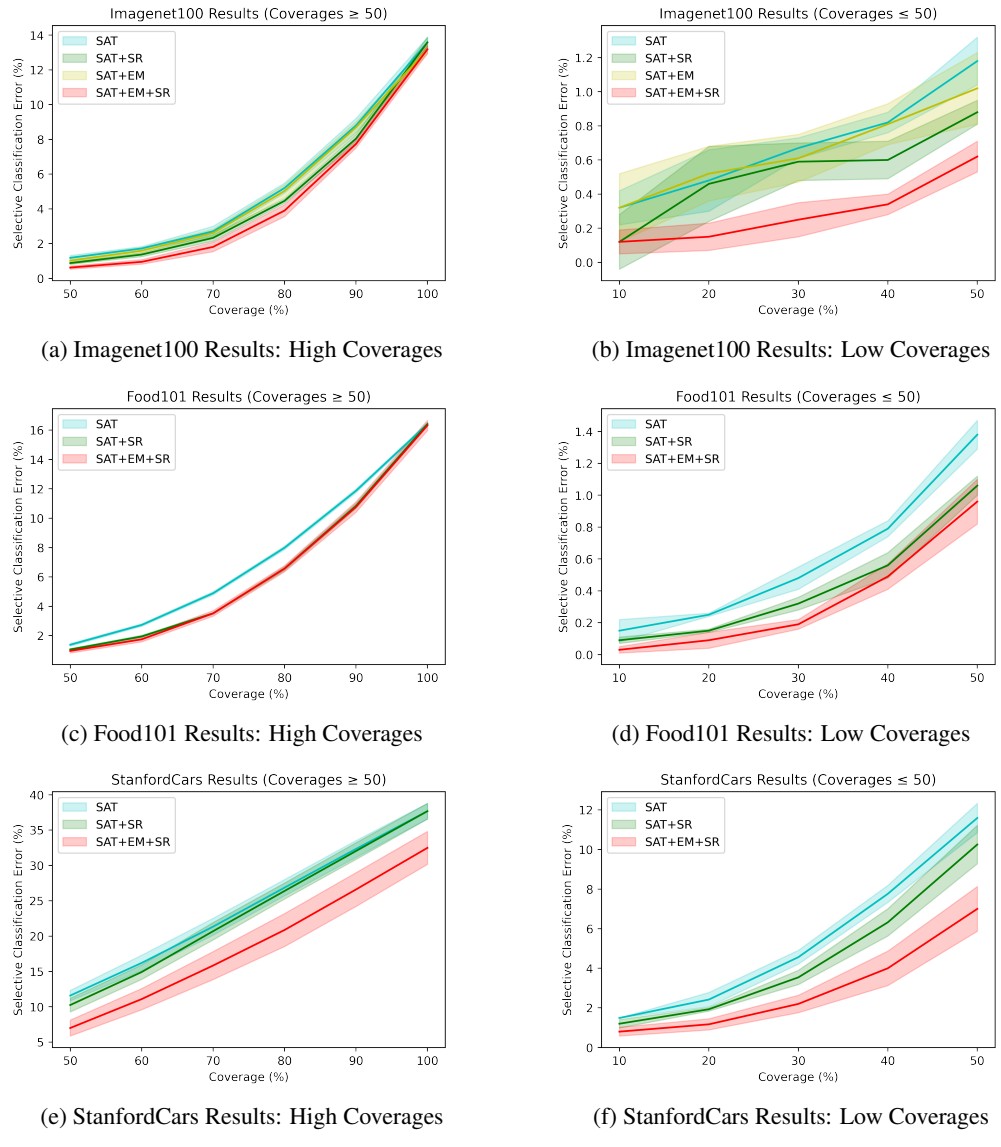

(a) Imagenet100 Results: High Coverages

(b) Imagenet100 Results: Low Coverages

(c) Food101 Results: High Coverages

(d) Food101 Results: Low Coverages

(e) StanfordCars Results: High Coverages

(f) StanfordCars Results: Low Coverages

Figure 7: Risk Coverage Plots for Imagenet10, Food101, and StanfordCars. All plots show that SAT+EM+SR outperform SAT across all coverages, achieving state-of-the-art results.

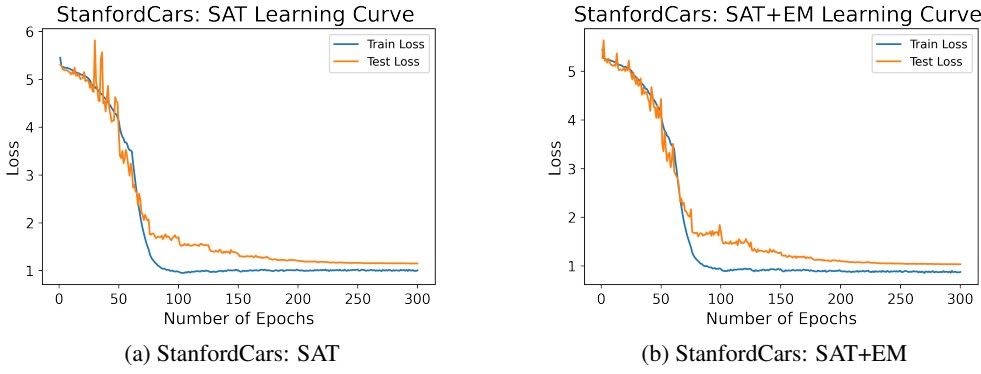

(a) StanfordCars: SAT

(b) StanfordCars: SAT+EM

Figure 8: Learning Curve (Convergence) Plots for StanfordCars.

