# OpenReview forum: "Towards Better Selective Classification"
_ICLR.cc/2023/Conference — ICLR 2023 poster_

### Official Review · Reviewer_6JTV · 2022-10-21

**Confidence:** 3
**Correctness:** 2
**Technical Novelty And Significance:** 2
**Empirical Novelty And Significance:** 2
**Recommendation:** 5

**Clarity, Quality, Novelty And Reproducibility:**

Clarity
* The idea is simple and easy to implement.
* Both bold and underlined values presented in all Tables are not properly explained. They should be explained in the regular paper, not only at the appendix.
* It is not clear how the results of Table 5 are obtained. The authors suggest the entropy-regularized loss function for semi-supervised purposes, but I did not find any mention about how they remove some labels of the supervised dataset.
* I am not sure how the authors select the coverage from the state-of-the-art methods. In the case of SelectiveNet, did the authors train the model choosing different $c_{target}$ values? In the case of the self-adaptive training, did the authors use the last Eq. of section 3.1? It seems the self-adaptive training is made with the purpose of discarding a solution only if the $C+1$ class has the highest accuracy.
* All equations should be numbered

Quality
* The provided results are interesting, but only one network was tested. I suggest the authors to introduce more networks with different architectures, in order to ensure the results provided are consistent.

Novelty
* Although the idea is simple, it could be very effective. However, I am not sure if the experimental results are fair, given the concern I had explain regarding to the clarity of the paper.
* Besides that, the entropy-regularization term for semi-supervised learning is not novel [1]. Although, as the entropy regularization is zero whenever all the values are o or 1, the idea is pretty similar to the active learning approach, where the non-labelled examples are labelled after each training epoch.

Reproducibility
* The authors provide enough information to successfully reproduce the networks, but they should include more information regarding how the state-of-the-art methods were trained (see the clarity section for details).

[1] Grandvalet, Y., & Bengio, Y. (2004). Semi-supervised learning by entropy minimization. Advances in neural information processing systems, 17.

**Strength And Weaknesses:**

Strength
* The idea is simple and easy to implement
* The results are interesting

Weaknesses
* The semi-supervised entropy regularization is not new [1].
* Only one network was tested.
* The paper organization is poor.

[1] Grandvalet, Y., & Bengio, Y. (2004). Semi-supervised learning by entropy minimization. Advances in neural information processing systems, 17.

**Summary Of The Paper:**

The authors propose to revise the conclusions of two different selective classification methods. They argue this methods obtain betters results by just providing a more robust training. Thus, at inference step, the selective solution can be discarded. With this variation, they are able to increase the results obtained by the original paper. Furthermore, they provide an entropy regularization term that can be used in semi-supervised learning.

**Summary Of The Review:**

Although the idea is interesting, the proposed solutions are either not new (entropy-regularization term) or not properly explained. Thus, I cannot recommend to publish this contribution in its actual form.

---

> ### Author Response · Authors · 2022-11-14
> **Response to Reviewer 6JTV (1/2)**
>
>
> > The semi-supervised entropy regularization is not new [1].
>
> Although the the idea of entropy regularization is not new in the semi-supervised learning literature, to the best of our knowledge, this is the first work that draws this connection to selective classification.
>
>
> > Only one network was tested.
>
> We have followed the same experimental setup as the previous selective classification works.
> In our experiments, we used two networks. For the CIFAR10 experiments, a VGG16 backbone was used for all methods. To accomodate for the larger amount of data in Imagenet-related experiments, Resnet34 was used. The paper is updated to clarify this information.
>
> To further address the reviwer's concern, we have included the results of using $3$ different architectures on a new dataset (StanfordCars) in the Experimental Results Section of the main comment. The results confirm our findings.
>
> > The paper organization is poor.
>
> If the description presented in the main comment of the rebuttal provides more clarity, we are happy to follow it in the paper presentation. Please let us know if you have any recommendations on how to improve the paper organization.
>
>
> > Both bold and underlined values presented in all Tables are not properly explained. They should be explained in the regular paper, not only at the appendix.
>
> Thanks for the suggestion! The paper is updated to include the explanation in the main paper.
>
> > It is not clear how the results of Table 5 are obtained. The authors suggest the entropy-regularized loss function for semi-supervised purposes, but I did not find any mention about how they remove some labels of the supervised dataset.
>
> We would like to clarify that no labels are removed from the dataset. Entropy regularization is applied on the predicted distribution by the network for all samples, so the labels are not needed for the regularization.
>
> We are minimizing the entropy of the predicted distribution which is equivalent to increasing the confidence of the model's predictions, benefitting the Softmax Response selection mechanism.
>
>
>
> > I am not sure how the authors select the coverage from the state-of-the-art methods. In the case of SelectiveNet, did the authors train the model choosing different $c_{target}$ values?
>
> For all selective classification methods, we followed the exact experimental setup as described in their papers. Specifically, this means for SelectiveNet, we trained a separate model for each $c_{target}$ value.
>
>
> > In the case of the self-adaptive training, did the authors use the last Eq. of section 3.1?
>
> There are two versions of the Self-Adaptive Training paper. The objective is Equation 12 in the IEEE Transactions [version](https://arxiv.org/pdf/2101.08732.pdf) and Equation 4 in the NeurIPS [version](https://arxiv.org/pdf/2002.10319.pdf).
>
> To ensure fairness in our experiments, we use the official public repo for self-adaptive training available [here](https://github.com/LayneH/SAT-selective-cls) and optimize the same objective. For experiments with the entropy-minimization objective, we simply add the term to the original Self-Adaptive Training objective.
>
>
>
> > All equations should be numbered
>
> Thank you for your suggestion, we've updated the paper to include the numbers.

---

> > ### Author Response · Authors · 2022-11-14
> > **Response to Reviewer 6JTV (2/2)**
> >
> >
> > > The provided results are interesting, but only one network was tested.
> >
> > > I suggest the authors to introduce more networks with different architectures, in order to ensure the results provided are consistent.
> >
> > Thank you for your suggestion. We followed the same experimental setup as the previous selective classification works. Our results are based on 2 architectures. Please check the main comment in the rebuttal for additional experiments in the Experimental Results Section including 2 additional datasets and various architectures.
> >
> >
> > > Although the idea is simple, it could be very effective. However, I am not sure if the experimental results are fair, given the concern I had explain regarding to the clarity of the paper.
> >
> > Please check the main comment in the rebuttal for more clarification on the experimental setup, contributions, datasets, and additional results.
> >
> > > Besides that, the entropy-regularization term for semi-supervised learning is not novel [1]. Although, as the entropy regularization is zero whenever all the values are o or 1, the idea is pretty similar to the active learning approach, where the non-labelled examples are labelled after each training epoch.
> >
> > In this case, we are not using entropy regularization for semi-supervised learning. Instead, we are drawing inspiration by showing a relationship between the two fields (Semi-supervised learning and selective classification) but applying the method to the selective classification (supervised learning) setting. We show that this simple idea significantly improves performance. The novelty is in the connection that we found and how we showed it improves performance in a non-semi-supervised learning setting.
> >
> >
> > > Although the idea is interesting, the proposed solutions are either not new (entropy-regularization term) or not properly explained. Thus, I cannot recommend to publish this contribution in its actual form.
> >
> > Please check the main comment in the rebuttal for more clarification on the experimental setup, contributions, datasets, additional architectures and additional results.
> >
> > We hope that the main comment helps clarify the importance of sharing our work with the community. If the description presented in the main comment of the rebuttal is preferred, we are happy to follow it in the paper presentation.

---

> ### Author Response · Authors · 2022-11-19
> **A Gentle Reminder to Reviewer 6JTV**
>
> Dear Reviewer 6JTV,
>
> We have addressed all of your concerns in the main comment of the rebuttal. Specifically, we have presented results on 2 additional datasets and 3 different architectures.
> Moreover, we have revised the results section of the main paper to enhance the presentation and highlight the strengths of our proposed methods.
>
> Could you please let us know if there are specific reasons you think this work is below the acceptance threshold?
> Are there any points that we failed to clarify?
> We are happy to address any doubts or questions you have and revise our paper accordingly. Any feedback would be highly appreciated.
> We look forward to hearing from you.

---

> ### Author Response · Authors · 2022-11-27
> **Kind Reminder to Reviewer 6JTV**
>
> We have addressed your concerns in the rebuttal:
>
>  - (1) **Experimental Fairness:** We described how the baselines followed their papers exactly (including using the official repositories), ensuring the fairness of our experiments.
>  - (2) **Lack of novelty due to applying entropy-minimization to a semi-supervised setting:** We clarified that we are not suggesting applying entropy-minimization to a semi-supervised setting. We are instead suggesting applying entropy-minimization to a **supervised setting**, specifically selective classification.
>
> Since we have addressed your concerns, are there any other specific reasons you think this work is below the acceptance threshold? We would highly appreciate any feedback. We are happy to address any other concerns you have.

---

> ### Author Response · Authors · 2022-12-03
> **Reminder to Reviewer 6JTV**
>
> Dear Reviewer 6JTV,
>
> We are sending you a reminder that the discussion period will be ending this month. Once again, we are happy to address any further concerns that you may have. We look forward to hearing from you.

---

> ### Author Response · Authors · 2022-12-08
> **Reminder to Reviewer 6JTV**
>
> Dear Reviewer 6JTV,
>
> We are sending you a reminder that the discussion period will be ending in a few days. We are happy to address any concerns that you may have. We look forward to hearing from you.

---

### Official Review · Reviewer_wKA4 · 2022-10-29

**Confidence:** 4
**Correctness:** 3
**Technical Novelty And Significance:** 2
**Empirical Novelty And Significance:** 2
**Recommendation:** 6

**Clarity, Quality, Novelty And Reproducibility:**

Authors make a useful observation for using selective classification, but whole idea of selective classification is to obtain a cost-sensitive prediction at training and inference time. As the current paper does not utilize the observed insight to propose a an improved selective classification training paradigm or provide more insights/practices, nor significant performance improvement.
Also, w.r.t entropy minimization regularization it might not be fruitful to minimize entropy on all instances?
Inspired by works which leverage uncertainty in case of pseudo labels[1] or even selective classification. In case of [2] authors discard hard samples to improve overall coverage vs accuracy tradeoff within selective classification.

References:
	1. M.N. Rizve et al . In Defence of Pseudo-Labeling: An Uncertainty aware pseudo label selection framewrok for semi-supervised learning (ICLR 2021)
	2. Y. Ding et al. Uncertainty-Aware Training of Neural Networks for Selective Medical Image Segmentation (MIDL 2020)


**Strength And Weaknesses:**

Strengths:

Paper is well written and easy to understand, and authors include various techniques of selective classification.

Weakness:

Though the point made is effective in experiments its only marginally better, including entropy regularized loss.
On CIFAR, we see the difference as we decrease coverage to 70 but even at 80% coverage selection mechanism and softmax response are at par and already <1% error. In case of imagenet we see 1.5% improvement at 80% coverage.

Also, authors do not provide further insight into why the observation exists, inspired by OOD detection work which demonstrate that max-logit is best performing OOD metric, and [3] makes an argument that max-logit is better because within Deep learning we check for familiarity of an input rather than absence of key features. So, based on OOD/anomaly detection signals & familiarity hypothesis we know it's always easy to detect presence than reject/ do good uncertainty quantification within our current deep learning practices.

Is there any reason for authors to not consider
	1. H. Mozannar et al. Consistent Estimators for Learning to Defer to an Expert (Learning to Defer) (ICML 2020)
	2. N. Charoenphakdee et al. Classification with Rejection Based on Cost-sensitive Classification (ICML 2021)
        3. T.G. Dietterich et al. The Familiarity Hypothesis: Explaining the Behavior of Deep Open Set Methods (arXiv: 2203.02486)

**Summary Of The Paper:**

Authors point out that predictive distribution base classifier within in selection classification captures
better signal for selection rather than a separate selection/abstention logit present in various approaches of selective classification within deep learning.


**Summary Of The Review:**

Though authors make an interesting point and demonstrate effectiveness of their observations. For lack of novelty, or lack of additional insights, currently in my humble opinion not convinced if the work warrants an ICLR acceptance.

---

> ### Author Response · Authors · 2022-11-14
> **Response to Reviewer wKA4 (1/2)**
>
>
> > Though the point made is effective in experiments its only marginally better, including entropy regularized loss. On CIFAR, we see the difference as we decrease coverage to 70 but even at 80% coverage selection mechanism and softmax response are at par and already <1% error.
> > In case of imagenet we see 1.5% improvement at 80% coverage.
>
> Please check the Dataset and the Experiment Results sections in the main comment of the rebuttal for a detailed description of the datasets and additional results on 2 more challenging datasets with various architectures.
>
> We believe that the error improvements by our EM+SR method are significant, especially at lower target coverages. For example, in table 5 of the main paper, for the target coverage of 50%, the SAT+EM+SR error is about half the SAT error. For less than 50% coverage, the improvement is doubled or tripled.
>
> Regarding the 1.5% improvement at 80%, we believe it is significant as well. Actually, it is representing ∼7 times the standard deviation ($\sigma=0.22$ at 80% coverage). Previous state-of-the-art work (Self-Adaptive Training) only improved upon SelectiveNet by 0.8% at 80% coverage. As such, our experimental results are significant. In previously published works at NeurIPS (Deep Gamblers and Self-Adaptive Training), the shown improvements on CIFAR-10 were even on the scale of 0.1%−0.2%.
>
> On StanfordCars, for $70$% coverage, using ResNet34 architecture, the selective classification is $21.34$%. For SAT+SR is $20.7$%. For $SAT+SR+EM$ is $15.84$%. In this case, the absolute percentage gain is $5.5$% and $25$% in relative gain.
>
>
> > Also, authors do not provide further insight into why the observation exists, inspired by OOD detection work which demonstrate that max-logit is best performing OOD metric, and [3] makes an argument that max-logit is better because within Deep learning we check for familiarity of an input rather than absence of key features. So, based on OOD/anomaly detection signals & familiarity hypothesis we know it's always easy to detect presence than reject/ do good uncertainty quantification within our current deep learning practices.
>
>
> We would like to thank the reviewer for drawing our attention to the work from [3]. This work is focusing on open set classification problem where the model can encounter new classes. We agree that the problems seem similar and the fact that max-logit achieves better performance supports our argument for using SR for selective classification. The connection between Selective Classification and OOD is very interesting but it is beyond the scope of this work.
>
> We would like to emphasize that the previous state-of-the-art methods (SN, DG, and SAT) are using special selection mechanisms; in this work, we show discarding the selection mechanism after training and using SR provides performance improvement at "no additional cost". Please check the main comment of the rebuttal for additional details.
>
>
>
> > Is there any reason for authors to not consider 1. H. Mozannar et al. Consistent Estimators for Learning to Defer to an Expert (Learning to Defer) (ICML 2020) 2. N. Charoenphakdee et al. Classification with Rejection Based on Cost-sensitive Classification (ICML 2021) 3. T.G. Dietterich et al. The Familiarity Hypothesis: Explaining the Behavior of Deep Open Set Methods (arXiv: 2203.02486)
>
> Thanks for the suggestion, we updated the text to cite [1], [2], and [3].
> - [1]: Is focusing on the classification in conjunction with expert decision makers. So for the discarded samples, the model does not just abstain, instead, the sample is passed to an expert where the expert has a certain accuracy.
> - [2]: This is an interesting paper about classification with cost-based rejection. The model abstains when the cost of rejection is less than the cost of misclassification. In this setup, there is no notion of target coverage. Consequently, the model design and experiments are completely different from our work as well as our baselines.
> - [3]: Please check the previous comment on OOD.
>
> We would like to highlight that our work is following the same lines as (SN, DG, and SAT) and none of these works ([1], [2], [3]) have provided a comparison to these top-performing selective classification methods.

---

> > ### Author Response · Authors · 2022-11-14
> > **Response to Reviewer wKA4 (2/2)**
> >
> >
> > > Authors make a useful observation for using selective classification, but whole idea of selective classification is to obtain a cost-sensitive prediction at training and inference time. As the current paper does not utilize the observed insight to propose a an improved selective classification training paradigm or provide more insights/practices, nor significant performance improvement.
> >
> > This is an interesting discussion regarding classification models with abstention mechanisms. However, there are two different setups for this problem. In the setting shared by SelectiveNet, Deep Gamblers, and Self-Adaptive Training, the setup is based on the target coverage. It means that for a given target coverage the empirical risk is minimized. The notion of risk and loss is similar to what is in cost-sensitive classification (e.g. N. Charoenphakdee et al). However, the introduction of the target coverage adds a new variable and changes the mathematical formulation. We refer the reviewer to the introduction and problem formulation in the SelectiveNet paper for more details. Regarding the results, we believe that our improvements are significant compared to the prior work as explained above and in the additional experiments presented in the main comment of the rebuttal.
> >
> > > Also, w.r.t entropy minimization regularization it might not be fruitful to minimize entropy on all instances?
> >
> > In our experiments, we tested different regularization weights. We found that entropy minimization with a small weight almost always improves the **overall** classification error on test samples. However, we agree that this might not be optimal for all the samples.
> >
> > An interesting future work can be to use instance-level weights for entropy regularization.
> >
> >
> >
> >
> > > Though authors make an interesting point and demonstrate effectiveness of their observations. For lack of novelty, or lack of additional insights, currently in my humble opinion not convinced if the work warrants an ICLR acceptance.
> >
> >
> > Please check the main section in the rebuttal for details on the contributions, novelty, datasets, and additional experiments on 2 challenging datasets and various architectures.
> >
> > We hope that the main comment helps clarify the importance of sharing our work with the community. If the description presented in the main comment of the rebuttal is preferred, we are happy to follow it in the paper presentation.

---

> > > ### Comment · Reviewer_wKA4 · 2022-11-23
> > > **Updating rating**
> > >
> > > I would like to thank authors for detailed feedback and updated experiments addressing many concerns!
> > >
> > > It is interesting to see entropy minimization gives further boost in selective classification, but could potentially have more bias as pointed by [1], I would encourage authors to perform some preliminary analysis w.r.t bias for broader community! To think from calibration perspective the updated classifier might have worse calibration but as selective classification addresses overall problem from different perspective we can ignore calibration.
> > >
> > > Overall paper highlights potential usage of selective classification methods, proposes entropy minimization to improve performance of selective classification but overall insights and novelty seem relatively marginal.
> > >
> > >
> > > References:
> > > 1. E. Jones et al. SELECTIVE CLASSIFICATION CAN MAGNIFY DISPARITIES ACROSS GROUPS (ICLR'21)

---

> > > > ### Author Response · Authors · 2022-11-27
> > > > **Thanking Reviewer wKA4**
> > > >
> > > > We would like to thank Reviewer wKA4 for their constructive feedback.
> > > >
> > > > Unlike [1] whose datasets were limited to settings with $2$ (binary classification) and $3$ classes, the datasets we are considering have a large number of $100+$ classes, making the analysis not trivial. As such, we believe that such analysis is deserving of a work on its own and is outside the scope of this specific paper. However, we definitely agree that such analysis would be interesting and we would be interested in studying this in future work.

---

> ### Author Response · Authors · 2022-11-19
> **A Gentle Reminder to Reviewer wKA4**
>
> Dear Reviewer wKA4,
>
> We would like to thank you for your detailed feedback.
> We have addressed all your concerns in the main comment of the rebuttal and in the comments below. Also, we have revised the paper and cited the suggested works. For the Results Section in the main paper, we have enhanced the presentation and added the additional datasets from the rebuttal to highlight the strong results of our proposed method.
>
>
> In the updated version, Table 2 shows the effect of discarding the original selection mechanism and using our softmax response (SR) selection mechanism for different selective models (SelectiveNet, Deep Gamblers,  and Sel-Adaptive Training). While Tables 3, 5, and 6 show that our proposed Softamx Response Selection with Entropy Minimization (EM) achieves new state-of-the-art selective classification results on very challenging datasets that were not considered by previous works.
>
>
> Could you please let us know if these changes have addressed your concerns? Are there other reasons you think this work should not be accepted?
> Please let us know if you need any further clarification. Any feedback would be highly appreciated. We look forward to hearing from you.

---

### Official Review · Reviewer_F9DQ · 2022-11-01

**Confidence:** 3
**Correctness:** 2
**Technical Novelty And Significance:** 2
**Empirical Novelty And Significance:** 2
**Recommendation:** 5

**Clarity, Quality, Novelty And Reproducibility:**

Clarity: The overall presentation is okay, but the description of the proposed method can be improved (cf. Summary of The Review)

Quality and Novelty: This aspect is average, as the proposed method is mainly based on entropy-regularized loss. The authors may need to clarify or highlight the novelty of the work.

Reproducibility: The results look reproducible.

**Strength And Weaknesses:**

**Strengths**

1. The authors introduced the state-of-the-art methods (SelectiveNet, Self-Adaptive Training, and Deep Gamblers) and the proposed method through mathematical expressions.
2. This paper analyzes the actual effective part of the state-of-the-art methods and the correlation between the Selective Classification problem and Semi-supervised learning. The proposed method has a better research premise and innovation.
3. The experiment compares the state-of-the-art methods with the proposed method and illustrates the point of view of the paper.

**Weakness**

1. The proposed method is a bit hard to follow.
2. The technique of the proposed method seems to be simple (i.e., mainly from the  entropy-regularized loss function).
3. The experiment is not well executed (e.g., $\beta$ is chosen as 0.01 for the proposed method, and not results under different settings).  The experimental results do not seem to work well on the Cifar10 dataset, while performing well on the two synthetic datasets generated by the authors. This kind of experimental results is unconvincing, as they may look artificial.

**Summary Of The Paper:**

The problem of Selective Classification considers how to train the model’s ability to abstain from a decision when the credibility is low. In this paper, the authors analyzed the state-of-the-art methods and found that learning a more generalizable classifier is more important than using some selection mechanisms in the problem of Selective classification. Besides, they found that the Softmax Response performs better than the entropy-based selection mechanism. Through researching the relationship between Selective Classification and Semi-supervised learning, the authors proposed an entropy-regularized loss function to improve the performance. The experimental results show the effectiveness of the proposed method.

**Summary Of The Review:**

* The authors may consider adding some visual expressions to help readers better understand the proposed method.
* The proposed method is mainly based on the  entropy-regularized loss function, which has a small amount of space in the article. The authors may point out more properites of the proposed method, so that readers can better appreciate the method. Besides, the authors may consider demonstrating the advantages of the proposed method compared with the state-of-the-art methods from the theoretical level.
* In the experiment, $\beta$ is chosen as 0.01 for the proposed method, the authors may consider giving some specific experiments to illustrate the effectiveness of the method under different settings.  The experiments do not seem to work well on the Cifar10 dataset, while the good results are on the two datasets constructed by the authors themselves. The authors need to add some datasets which also used by other researchers to improve the credibility of the experiment.

* Typo: the the -> the (cf. the last paragraph line 4 on page 4)

---

> ### Author Response · Authors · 2022-11-14
> **Response to Reviewer F9DQ**
>
>
> We would like to thank the reviewer for their feedback.
>
> > The proposed method is a bit hard to follow.
>
> Please refer to the main comment of the rebuttal for more details on the contribution, datasets, and experimental results.
>
> > The technique of the proposed method seems to be simple (i.e., mainly from the entropy-regularized loss function).
>
> Please check the Contributions section in the main comment of the rebuttal and the experimental results section where we have added 2 additional datasets and several architectures that further demonstrates the gains of our method.
>
> > The experimental results do not seem to work well on the Cifar10 dataset, while performing well on the two synthetic datasets generated by the authors. This kind of experimental results is unconvincing, as they may look artificial.
>
> > The authors need to add some datasets
>
> Similar to previous works on selective classification, we have included CIFAR10. However, as we mentioned in the paper, this dataset is already saturated. On the contrary, our datasets are more challenging. We would like to clarify that the datasets Imagenet100 and ImageNetSubset are not synthetic.
>
> Please check the Datasets section in the main comment of the rebuttal for more details. To further demonstrate the strength of our experiments, we have included results on 2 additional datasets with various architectures and clarified the gains for our proposed methods in the Experiment Results section in the main comment.
>
>
> > Quality and Novelty: This aspect is average, as the proposed method is mainly based on entropy-regularized loss. The authors may need to clarify or highlight the novelty of the work.
>
> Please check the Contributions section in the main comment of the rebuttal.
>
> We hope that the main comment helps clarify the importance of sharing our work with the community. If the description presented in the main comment of the rebuttal is preferred, we are happy to follow it in the paper presentation.
>
> > The proposed method is mainly based on the entropy-regularized loss function, which has a small amount of space in the article. The authors may point out more properties of the proposed method, so that readers can better appreciate the method.
>
>
> Please check the Contributions section in the main comment of the rebuttal.
> An important message from this work is to correct a misconception about the selection mechanism. Specifically, we propose to "[d]iscard the [original] selection mechanism and use Softmax Response (SR) for selection on a pretrained selective classifier". This provides clear gains without any additional costs.
>
> Furthermore, the performance gains are not only due to entropy regularized loss function.  In fact, Table 6 shows that using entropy minimization (EM) (with the original SAT selection mechanism) alone only provides slight improvement over SAT. The large performance gains come from using SR alongside EM.
>
>
>
>
> > In the experiment, $\beta$ is chosen as 0.01 for the proposed method, the authors may consider giving some specific experiments to illustrate the effectiveness of the method under different settings.
>
> For $\beta$, we have tried 3 values on ImageNet100 (0.001, 0.01, and 0.1) on a validation set. The best value was used for all the other datasets. We will clarify the details in the supplementary material.
>
> > The experiments do not seem to work well on the Cifar10 dataset, while the good results are on the two datasets constructed by the authors themselves. The authors need to add some datasets which also used by other researchers to improve the credibility of the experiment.
>
> Please check the Dataset and the Experiment Results sections in the main comment of the rebuttal for a detailed description of the datasets and additional results on two new datasets and various architectures. We would like to clarify that Imagenet100 was not constructed by ourselves.
>
>
>
> > Typo: the the -> the (cf. the last paragraph line 4 on page 4)
>
> Thanks, we have fixed it.

---

> ### Author Response · Authors · 2022-11-19
> **A Gentle Reminder to Reviewer F9DQ**
>
> Dear Reviewer F9DQ,
>
> We have addressed all of your concerns in the main comment of the rebuttal and in the comment below. We have highlighted the contributions, dataset details, and additional experimental results with 2 new datasets and several architectures.  Finally, we have revised the results section of the main paper to enhance the presentation, clarify hyperparameter choices, and highlight the strengths of our proposed method.
>
>
> Could you please let us know if there are other reasons you think this work is below the acceptance threshold?
> Please let us know if you have any further questions. Any feedback would be highly appreciated.
> We look forward to hearing from you.

---

> ### Author Response · Authors · 2022-11-27
> **Kind Reminder to Reviewer F9DQ**
>
> We have addressed your concerns in the rebuttal:
>
>  - (1) **lack of datasets:** We have included results on $2$ additional datasets (for a total of $5$ datasets!), further showing our proposed method significantly outperforms the previous state-of-the-art.
>  - (2) **a bit hard to follow:** To clarify the benefits of the work, we have revised the results section and included the results on the additional datasets.
>
> Since we have addressed your concerns, are there any other specific reasons you think this work is below the acceptance threshold? We would highly appreciate any feedback. We are happy to address any other concerns you have.

---

> ### Author Response · Authors · 2022-12-05
> **Reminder to Reviewer F9DQ**
>
> Dear Reviewer F9DQ,
>
> We are sending you a reminder that the discussion period will be ending in a week. Once again, we are happy to address any further concerns that you may have. We look forward to hearing from you.

---

> ### Author Response · Authors · 2022-12-09
> **Reminder to Reviewer F9DQ**
>
> Dear Reviewer F9DQ,
>
> We are sending you a reminder that the discussion period will be ending in $3$ days. In our rebuttal, we have addressed your concerns, including running experiments on new datasets, further highlighting the significance of our results. We are more than happy to address any further concerns that you may have. We look forward to hearing from you.

---

### Official Review · Reviewer_hDSo · 2022-11-01

**Confidence:** 4
**Correctness:** 3
**Technical Novelty And Significance:** 3
**Empirical Novelty And Significance:** 3
**Recommendation:** 8

**Clarity, Quality, Novelty And Reproducibility:**

I found the paper well written. The experiments were thorough and the relation with semi-supervised learning is indeed novel, albeit less well fleshed out.

**Strength And Weaknesses:**

Pros:
I found the paper to be generally very well written and the experiments well motivated. The most interesting part of the paper to me was the connection with self-supervised models.
I also appreciated the authors reporting the results with errorbar statistics. I wish more CS/AI researchers emulate this
I found the results to be quite thorough. I liked seeing the results generalize to different tests and reported across the number of classes (where selective decision becomes more of a problem).

Cons:
In some instances, I felt the results were slightly overstated. The difference between the entropy loss and the SR loss is relatively subtle across the board.
The relation with semi-supervised learning-based models could be better explored. At the moment it is an observation. Readers would like to get an intuition for why this happens.


**Summary Of The Paper:**

The paper tackles an interesting question regarding model decision making. Specifially, the authors are looking into the ability of a model to abstain from making a decision, when it has low confidence scores: The “Selective Classification problem”.

The authors demonstrate that:
1) the current SOA models’s performance can be attributed to their ability to train a more classifier than is more generalizable (than the logit selection mechanism).
2) semi-supervised learning-based models have better selective classification ability.
3) the new selective classifier (SR: SoftMax response) could improve current SOA models


**Summary Of The Review:**

Overall the paper addresses an important question about Selective Classification. I was wondering if there are practical applications of this specific decision making approach. It would be a good addition to articulate the utility of this approach, especially in CV related tasks. The relation with the semi-supervised learning approaches is also interesting but less well understood. I encourage the authors to spend more time fleshing that section out in the paper (or in future work).

---

> ### Author Response · Authors · 2022-11-14
> **Response to Reviewer hDSo**
>
>
> We would like to thank the reviewer for their very supportive feedback. We are pleased to see your enthusiasm for our work.
>
>
>
> > Cons: In some instances, I felt the results were slightly overstated. The difference between the entropy loss and the SR loss is relatively subtle across the board. The relation with semi-supervised learning-based models could be better explored. At the moment it is an observation. Readers would like to get an intuition for why this happens.
>
> Please check the Experiment Results section in the main comment of the rebuttal. We have included 2 additional datasets with various architectures and clarified the clear gains of using SR and SR+EM. As for the relation to semi-supervised learning, we are hoping that our work would open the door for more exploration in this area.

---

### Author Response · Authors · 2022-11-14
**Main Comment: Experiment Results**

## Comparison of Selective Classification Methods
In the following table, we include a comparison on sample target coverages between existing baselines and our proposed methods: Softmax Response on the pretrained selective classifier (SR) and the use of an entropy minimization regularizer (EM). The numbers represent the selective classification error. The target coverage is shown in parentheses alongside the name of the dataset.

|      Dataset\Method      | SelectiveNet |      DG     |     SAT     |    SAT+SR   |    SAT+SR+EM    |
|:------------------------:|:------------:|:-----------:|:-----------:|:-----------:|:---------------:|
|       CIFAR10 (90%)      |  2.49 ± 0.13 |  2.27 ± 0.0 | 2.18 ± 0.11 | 2.11 ± 0.06 | **2.16 ± 0.01** |
|    ImageNet-100 (80%)    |  6.00 ± 0.2  |  5.21 ± 0.3 |  5.20 ± 0.3 | 4.46 ± 0.13 |  **3.9 ± 0.34** |
| ImageNetSubset-150 (70%) |  3.68 ± 0.27 | 2.62 ± 0.03 | 2.23 ± 0.16 | 1.89 ± 0.15 | **1.71 ± 0.15** |


## StanfordCars Dataset:
 The following table presents the selective classification error on StanfordCars. Since SAT is the best baseline for selective classification, we include SAT, SAT+SR (to show the effect of changing the selection mechanism to SR), and SAT+SR+EM. As requested by reviewer 6JTV, we have included 3 architectures ResNet34, RegNetX, and ShuffleNet.
|     |   ResNet34   |   ResNet34   |     ResNet34     |   |    RegNetX   |    RegNetX   |      RegNetX     |   |  ShuffleNet  |  ShuffleNet  |    ShuffleNet    |
|:---:|:------------:|:------------:|:----------------:|:-:|:------------:|:------------:|:----------------:|:-:|:------------:|:------------:|:----------------:|
|     |      SAT     |    SAT+SR    |     SAT+EM+SR    |   |      SAT     |    SAT+SR    |     SAT+EM+SR    |   |      SAT     |    SAT+SR    |     SAT+EM+SR    |
| 100 | 37.68 ± 1.11 | 37.68 ± 1.11 | **32.49 ± 2.33** |   | 31.78 ± 2.44 | 31.78 ± 2.44 | **27.75 ± 1.81** |   | 34.10 ± 0.73 | 34.10 ± 0.73 | **32.90 ± 1.29** |
|  90 | 32.34 ± 1.19 | 32.04 ± 1.18 | **26.60 ± 2.39** |   | 26.35 ± 2.43 | 25.68 ± 2.44 | **21.72 ± 1.90** |   | 28.61 ± 0.72 | 28.27 ± 0.80 | **26.94 ± 1.33** |
|  80 | 26.86 ± 1.15 | 26.39 ± 1.13 | **20.87 ± 2.33** |   | 21.20 ± 2.40 | 20.07 ± 2.54 | **16.21 ± 1.79** |   | 23.16 ± 0.47 | 22.72 ± 0.63 | **21.13 ± 1.40** |
|  70 | 21.34 ± 1.20 | 20.70 ± 1.23 | **15.84 ± 1.98** |   | 16.45 ± 2.14 | 14.77 ± 2.23 | **11.22 ± 1.54** |   | 17.94 ± 0.27 | 17.14 ± 0.46 | **15.70 ± 1.42** |
|  60 | 16.21 ± 1.10 | 14.92 ± 1.03 | **11.09 ± 1.50** |   | 12.13 ± 1.64 | 10.07 ± 1.58 |  **7.39 ± 1.21** |   | 13.00 ± 0.24 | 12.10 ± 0.46 | **10.89 ± 1.19** |
|  50 | 11.59 ± 0.74 | 10.25 ± 0.97 |  **7.00 ± 1.13** |   |  8.60 ± 1.27 |  6.43 ± 1.46 |  **4.55 ± 0.96** |   |  9.23 ± 0.10 |  7.68 ± 0.10 |  **7.11 ± 0.87** |
|  40 |  7.76 ± 0.43 |  6.32 ± 0.69 |  **4.00 ± 0.87** |   |  5.94 ± 1.06 |  4.04 ± 0.88 |  **2.88 ± 0.61** |   |  6.31 ± 0.22 |  4.77 ± 0.24 |  **4.49 ± 0.51** |
|  30 |  4.56 ± 0.35 |  3.54 ± 0.36 |  **2.20 ± 0.44** |   |  3.99 ± 0.60 |  2.47 ± 0.44 |  **1.74 ± 0.34** |   |  3.81 ± 0.39 |  2.97 ± 0.25 |  **2.83 ± 0.28** |
|  20 |  2.42 ± 0.36 |  1.93 ± 0.09 |  **1.17 ± 0.28** |   |  2.55 ± 0.33 |  1.55 ± 0.00 |  **1.10 ± 0.34** |   |  2.07 ± 0.34 |  1.70 ± 0.30 |  **1.43 ± 0.05** |
|  10 |  1.49 ± 0.00 |  1.20 ± 0.21 |  **0.80 ± 0.22** |   |  1.66 ± 0.26 |  0.91 ± 0.15 |  **0.70 ± 0.26** |   |  1.08 ± 0.26 |  0.99 ± 0.18 |  **0.66 ± 0.31** |

## Food101 Dataset:
  The following table presents the selective classification error on Food101
|     |      SAT     |    SAT+SR    |     SAT+EM+SR    |
|:---:|:------------:|:------------:|:----------------:|
| 100 | 16.41 ± 0.10 | 16.41 ± 0.10 | **16.32 ± 0.35** |
|  90 | 11.87 ± 0.13 | 10.84 ± 0.17 | **10.77 ± 0.36** |
|  80 |  7.99 ± 0.12 |  6.57 ± 0.13 |  **6.57 ± 0.21** |
|  70 |  4.89 ± 0.11 |  3.52 ± 0.05 |  **3.52 ± 0.19** |
|  60 |  2.73 ± 0.09 |  1.95 ± 0.08 |  **1.75 ± 0.17** |
|  50 |  1.38 ± 0.09 |  1.06 ± 0.06 |  **0.96 ± 0.14** |
|  40 |  0.79 ± 0.05 |  0.56 ± 0.08 |  **0.49 ± 0.08** |
|  30 |  0.48 ± 0.07 |  0.32 ± 0.04 |  **0.19 ± 0.03** |
|  20 |  0.25 ± 0.01 |  0.15 ± 0.01 |  **0.09 ± 0.05** |
|  10 |  0.15 ± 0.07 |  0.09 ± 0.02 |  **0.03 ± 0.02** |


## Conclusions on the experiments
From the experiment results on 5 datasets (3 in the paper, and 2 in the rebuttal), it is clear that changing the selection mechanism to SR provides performance gains over SAT (the previous best selective classification model) without any additional costs. Moreover, our proposed entropy-minimization (EM) term when combined with SR, provides new state-of-the art selective classification results. This confirms our reported findings in the paper. It is important to note that the gains of our proposed methods are consistent across different architectures and different datasets with different numbers of classes as demonstrated by the additional experiments presented in the rebuttal.

---

### Author Response · Authors · 2022-11-14
**Main Comment: Datasets**


The recent works on selective classification have been summarized in the following table with their experimental datasets:

| Paper                        | Conference/Journal                   | Datasets                       | Selection Mechanism |
|------------------------------|--------------------------------------|--------------------------------|---------------------|
| SelectiveNet (SN)            | ICML 2019                            | CIFAR-10 and Cats vs. Dogs     | Selection Head      |
| Deep Gamblers (DG)           | NeurIPS 2019                         | SVHN, CIFAR10 and Cat vs. Dogs | Abstain Logit       |
| Self-Adaptive Training (SAT) | NeurIPS 2020, IEEE Transactions 2022 | SVHN, CIFAR10 and Cat vs. Dogs | Abstain Logit       |


We would like to highlight that all the previous selective classification methods have focused only on the following datasets: CIFAR10, Cats vs. Dogs and SVHN. The problem with these datasets is that the error quickly goes to zero as the target coverage decreases. To show this, the following table
provides the classification error of Self-Adaptive Training on these 3 datasets as well as our 4 additional datasets: ImageNet-100, ImagenetSubset, StanfordCars, and Food101.



| Dataset\Coverage                    | 100% | 90%  | 80%  | 70%  |
|-------------------------------------|------|------|------|------|
| CIFAR10                             | 6%   | 2.2% | 0.7% | 0.3% |
| Cats vs. Dogs                       | 3%   | 0.6% | 0.2% | 0%   |
| SVHN                                | 3%   | 0.6% | 0.4% | 0%   |
| ImageNet-100 (Ours)                 | 14%  | 9.4% | 6%   | 3.4% |
| ImagenetSubset (175 Classes) (Ours) | 15%  | 9.7% | 5.4% | 2.7% |
| StanfordCars (Ours)                 | 38%  | 32%  | 27%  | 21%  |
| Food101 (Ours)                      | 16%  | 12%  | 8%   | 5%   |


The error at 80% coverage is already lower than 1% for CIFAR10, Cats vs. Dogs and SVHN. Since the possible margin of improvement is not substantial, it's difficult to draw meaningful conclusions about the performance of different methods. As such, in this work, we introduce more challenging datasets and benchmark the existing state-of-the-art methods:

- **Imagenet100** which was initially used by Tian et al., (2020) for classification and not selective classification. The difficulty of Imagenet-100 enables us to compare different methods at coverages as low as 10%.
- **ImagenetSubset** which varies the number of classes from 25 to 175. The goal of this dataset is to compare the performance of the different methods as the number of classes increases. Previous datasets were limited to at most 10 classes.

Following the recommendation by the reviewers, we are including 2 additional datasets in the rebuttal. Similar to our ImageNet datasets, it is clear from the previous table that these datasets are not saturated:
- **StanfordCars**: The dataset is available [here](https://ai.stanford.edu/~jkrause/cars/car_dataset.html). The Cars dataset contains 8,144 training images and 8,041 testing images split into 196 classes of cars. We ran the experiment for 300 epochs.
- **Food101**:The dataset is available [here](https://data.vision.ee.ethz.ch/cvl/datasets_extra/food-101/). The Food dataset contains 75750 training images and 25250 testing images split into 101 food categories. We trained the model for 500 epochs.

---

### Author Response · Authors · 2022-11-14
**Main Comment: Contributions**

We would like to highlight that our main contributions are:
 - **(1) Selection Mechanism:**
We present a surprising finding that discarding the selection mechanism of a pretrained selective classifier, and using the classic softmax response (SR) instead, improves the selective classification performance "*at no additional cost*". This surprising observation applies to the three main selective classification methods: SelectiveNet, Deep Gamblers, and SAT. We argue that this observation is important for the community to encourage the better design of selective models and prevent the same shortcoming in future works.
 - **(2) Entropy Minimization:**
 We draw inspiration from semi-supervised learning and show that entropy-minimization regularization, a common technique used in semi-supervised learning, significantly improves the performance of the state-of-the-art selective classification methods. To the best of our knowledge, this is the first work that has proposed this connection in selective classification.
 - **(3) Datasets:**
This is the first work that expands the empirical evaluation for selective classification beyond low-resolution saturated datasets and studies the effect of the number of classes on performance. We will expand more on this point in the following section.
- **(4) Results:**
Our proposed method using both Softmax Response selection and Entropy minimization achieves new state-of-the-art selective classification results on 4 challenging datasets (2 in the main paper and 2 in the rebuttal). The performance gains are consistent across various architectures and number of classes.

---

### Author Response · Authors · 2022-11-18
**Discussion Phase Summary**

Dear Reviewers,
As the discussion phase is ending in a few ending in a few hours, we wanted to summarize the discussion.
The reviewers' concerns were mostly about the datasets, number of architectures used, and novelty. In the main comment of the rebuttal, we have highlighted the contributions, dataset details, and additional experimental results with 2 new datasets and several architectures. Additionally, based on the reviewers' feedback, we have enhanced the results presentation in the main paper to highlight the strengths of our proposed method.

Our work (1) corrects a misconception within the community on selection mechanisms, showing how existing methods are ineffective (2) is the first to highlight the connection between selective classification and semi-supervised learning and (3) achieves state-of-the-art results by a large margin on several challenging unsaturated datasets. For the aforementioned three reasons, we believe this work is practically important and useful to the community.

Please let us know if you have any further comments.

---

### Decision · Program_Chairs · 2023-01-20

**Decision:**

Accept: poster

**Justification For Why Not Higher Score:**

There are still some remaining issues.

**Justification For Why Not Lower Score:**

N/A

**Metareview: Summary, Strengths And Weaknesses:**

The paper main contribution is the finding that selective classification performance can be boosted by simply using softmax response (SR) on a network that was pretrained with a selection mechanism. The authors consider to the three main selective classification methods: SelectiveNet, Deep Gamblers, and SAT. It is also shown that entropy minimization regularization improves the selective classification performance.
I believe this paper contains interesting and valuable observations and recommend acceptance.
I ask the authors to address the following issues:

1) An analysis over the complete ImageNet dataset would greatly benefit this paper. ImageNet results are highly transferable to real-world, while results on subsets of ImageNet do not always transfer even to the entire ImageNet dataset itself. Moreover, a random selection of a dataset subset can be problematic, unless it is repeated several times.

2) The paper could be streamlined, and while it is good to provide hypotheses that explain some observations, I feel some are not based on enough evidence (such as the claims about generalization).

3) A risk-coverage figure depicting the different methods could greatly improve the comparison (the tables with selective risks for different coverages are okay, but not as informative). If you want to compare only higher coverages, you can start the curves at 50% coverage, for example.

4) In some of the tables (e.g., Table 3) the boldface highlighting of SAT+EM+SR is not correct in all cases. In case of an overlap It would probably be fair to emphasize the entry with the lower standard error or all the overlapping confidence intervals.

I look forward to seeing your paper published once you fix these issues!


**Note From Pc:**

if the above contains the word "oral" or "spotlight" please see: "oral" presentation means -> notable-top-5% and "spotlight" means -> notable-top-25%. As stated in our emails, we are disassociating presentation type from AC recommendations